# Type II alveolar cell MHCII improves respiratory viral disease outcomes while exhibiting limited antigen presentation

Sushila A. Toulmin [1,2✉], Chaitali Bhadiadra[1], Andrew J. Paris [3], Jeffrey H. Lin [2], Jeremy Katzen[3], Maria C. Basil[4], Edward E. Morrisey[4,5,6,7], G. Scott Worthen[8,9] & Laurence C. Eisenlohr [1,10✉]

Type II alveolar cells (AT2s) are critical for basic respiratory homeostasis and tissue repair after lung injury. Prior studies indicate that AT2s also express major histocompatibility complex class II (MHCII) molecules, but how MHCII expression by AT2s is regulated and how it contributes to host defense remain unclear. Here we show that AT2s express high levels of MHCII independent of conventional inflammatory stimuli, and that selective loss of MHCII from AT2s in mice results in modest worsening of respiratory virus disease following influenza and Sendai virus infections. We also find that AT2s exhibit MHCII presentation capacity that is substantially limited compared to professional antigen presenting cells. The combination of constitutive MHCII expression and restrained antigen presentation may position AT2s to contribute to lung adaptive immune responses in a measured fashion, without over-amplifying damaging inflammation.

[1] Division of Protective Immunity, Department of Pathology and Laboratory Medicine, Children's Hospital of Philadelphia, Philadelphia, PA, USA. [2] Perelman School of Medicine, University of Pennsylvania, Philadelphia, PA, USA. [3] Division of Pulmonary, Allergy and Critical Care Medicine, Department of Medicine, Perelman School of Medicine, University of Pennsylvania, Philadelphia, PA, USA. [4] Department of Medicine, Penn-CHOP Lung Biology Institute, University of Pennsylvania, Philadelphia, PA, USA. [5] Penn Cardiovascular Institute, Perelman School of Medicine, University of Pennsylvania, Philadelphia, PA, USA. [6] Penn Institute for Regenerative Medicine, Perelman School of Medicine, Philadelphia, PA, USA. [7] Department of Cell and Developmental Biology, Perelman School of Medicine, University of Pennsylvania, Philadelphia, PA, USA. [8] Department of Pediatrics, University of Pennsylvania Perelman School of Medicine, Philadelphia, PA, USA. [9] Division of Neonatology, Children's Hospital of Philadelphia, Philadelphia, PA, USA. [10] Department of Pathology and Laboratory Medicine, Perelman School of Medicine, University of Pennsylvania, Philadelphia, PA, USA. ✉email: sushila.toulmin@pennmedicine.upenn.edu; eisenlc@pennmedicine.upenn.edu

CD4[+] T cell responses are initiated by T cell receptor (TCR) recognition of cognate major histocompatibility complex class II (MHCII)/peptide complexes on the surface of antigen-presenting cells (APCs). By convention, constitutive MHCII expression is restricted to a few specialized types of cells, including the "professional APCs"—B cells, dendritic cells (DCs), and macrophages—as well as thymic epithelial cells (TECs)[1]. In contrast, all other cell types are thought to produce MHCII only in the setting of inflammation, specifically in response to the cytokine interferon-γ (IFNγ)[2]. However, more recently, an increasing number of other cell types have been shown to express MHCII at homeostasis and to contribute to adaptive immune responses in a variety of settings[3,4]. These include certain intestinal ILC3 subsets[5,6], which express MHCII independent of IFNγ, as well as several nonhematopoietic cell types, such as lymph node stromal cells[7–10], vascular endothelial cells[11–13], and intestinal epithelial cells[14–16], whose MHCII production is driven by the basal IFNγ present at steady state.

Type II alveolar cells (AT2s) are an additional non-hematopoietic cell type that have been reported to express MHCII at homeostasis in both rodents[17–22] and humans[23–27]. AT2s are epithelial cells present in the lung parenchyma, and their main physiologic functions are to produce surfactant and to serve as stem-like cells in the distal lung[28–31]. AT2s have also been described to perform several innate immunologic functions[32–34], and their constitutive MHCII expression suggests that they may participate in lung adaptive immune responses as well. Previous studies of AT2 MHCII function have varied widely, with some concluding that AT2s serve to activate CD4[+] T cells[19,21] and others demonstrating the opposite[20,26]. As a result, both the MHCII presentation capability of AT2s and the contribution of AT2 MHCII to lung host responses remain unclear. Here we report on the mechanisms regulating homeostatic MHCII expression in AT2s, their antigen presentation capacity, and the impact of AT2 MHCII in vivo.

## Results

### AT2s express MHCII protein at homeostasis to uniformly high levels.
To verify prior reports of homeostatic AT2 MHCII expression, we assessed MHCII on both mouse and human AT2s by flow cytometry. We found that naive wild-type C57Bl/6 (B6) AT2s, identified as CD45[−]CD31[−]EpCAM[int] (Supplementary Fig. 1)[22], uniformly express MHCII (Fig. 1a). Similarly, healthy human HT2-280[+] AT2s[35] express the human MHCII protein HLA-DR (Supplementary Fig. 2a, b).

We next compared MHCII levels on AT2s to other well-studied MHCII-expressing lung cells in naive mice. AT2s express MHCII at a magnitude similar to that of lung professional APCs, dendritic cells (DCs) and B cells, and higher than alveolar macrophages (Fig. 1b). Besides AT2s, vascular endothelial cells (ECs) are another nonhematopoietic cell type in the lung that express MHCII at homeostasis[36]. Compared to ECs, AT2 MHCII expression is higher and more homogeneous (Fig. 1c).

MHCII protein may be synthesized intracellularly or extrinsically acquired[37] from conventional APCs. We examined the source of AT2 MHCII by RNA-sequencing analysis of naive mouse AT2s [data originally generated by Ma et al.[38]], qPCR, and by creating B6 CD45.1 MHCII[+/+] (B6[CD45.1]) and CD45.2 MHCII[−/−] (MHCII[−/−]) bone marrow chimeric mice. AT2s express transcript for both the α and β chains of the MHCII allele in B6 mice, I-A[b] (Fig. 1d, Supplementary Fig. 5a), and AT2s retained MHCII protein expression to high levels in MHCII[−/−] → B6[CD45.1] mice (Fig. 1e). Thus, AT2s synthesize their own MHCII and do not depend on acquisition of MHCII protein from hematopoietic APCs.

In sum, AT2s represent a unique population of nonhematopoietic lung cells that produce MHCII protein at homeostasis uniformly to high levels, similar to that of professional APCs.

### AT2 MHCII expression is regulated in an unconventional, IFNγ-independent manner.
The class II transactivator (CIITA) is the master transcription factor that drives MHCII expression[2]. To determine whether AT2 MHCII expression is dependent on CIITA, we assessed MHCII in the lungs of Ciita[−/−] mice. Lack of CIITA results in the complete loss of MHCII expression on AT2s, as well as lung DCs and ECs (Fig. 1f). Transcription of CIITA begins at three different promoter sites (pI, pIII, and pIV) depending on cell and context specific stimuli. Recruitment of pI occurs in DCs and macrophages, pIII in B cells and plasmacytoid DCs, and pIV in ILC3s and nonhematopoietic cells[2,6]. To assess whether AT2 MHCII is dependent on pIV, we evaluated mice lacking just the pIV region within the CIITA locus[39]. AT2s from Ciita pIV[−/−] mice fail to express MHCII, as do ECs, while DCs retain MHCII expression as expected (Fig. 1f).

The recruitment of pIV in all nonhematopoietic cells (except for thymic epithelial cells (TECs)) is thought to require induction by the cytokine IFNγ[2]. We evaluated MHCII in mice deficient in IFNγ, IFNγR, and the associated downstream pIV-binding transcription factor, STAT1. AT2s retain MHCII expression in Ifng[−/−], Ifngr1[−/−], and Stat1[−/−] mice in a manner similar to DCs and in contrast to ECs, whose MHCII expression is abrogated in the absence of IFNγ signaling (Fig. 1g). AT2s, like DCs, exhibit a small reduction in MHCII expression in Stat1[−/−] mice, indicating that IFNγ signaling may upregulate MHCII expression in AT2s, but it is not required for steady state expression.

Besides IFNγ, no other cytokine has been shown to be sufficient to induce MHCII expression in nonhematopoietic cells. However, given the IFNγ-independent expression of MHCII by AT2s, we also examined the requirement of a broad array of conventional innate and adaptive immune mediators, by assessing Ifnar1[−/−], Ifnar1[−/−]Ifngr1[−/−], Myd88[−/−], Stat6[−/−], and germ-free mice. AT2s express MHCII at wild-type levels in all cases (Supplementary Fig. 3), indicating that they also do not require a wide spectrum of other, non-IFNγ inflammatory signals for MHCII expression, including type I IFNs, TLR ligands, IL-1, IL-18, IL-33, Th2 cytokines, and the microbiota.

Altogether, these data demonstrate that AT2 homeostatic MHCII expression is CIITA pIV-dependent, but IFNγ-independent. This same transcriptional configuration has been reported for only two other cell types: TECs[40], which are present in a primary immune organ, and ILC3s[6], which are immune cells. To our knowledge it has never been described for a cell outside of the immune system.

### AT2s possess classical APC characteristics and MHCII antigen processing and presentation machinery.
We next examined whether AT2s possess mediators of classical MHCII presentation. By convention, this pathway begins with extracellular antigen acquisition followed by proteolysis in the endocytic compartment[41–43]. To assess these steps, we evaluated the capacity of AT2s to catabolize DQ-Ova, a self-quenched conjugate of ovalbumin protein that fluoresces after receptor-mediated endocytosis and proteolytic digestion. AT2s exhibited a temperature-dependent increase in fluorescence when incubated with DQ-Ova, less so than DCs but similarly to B cells, indicating that they are capable of active antigen uptake and degradation (Fig. 2a). Antigen degradation is mediated by several key enzymes, including asparagine endopeptidase (AEP), gamma-interferon-

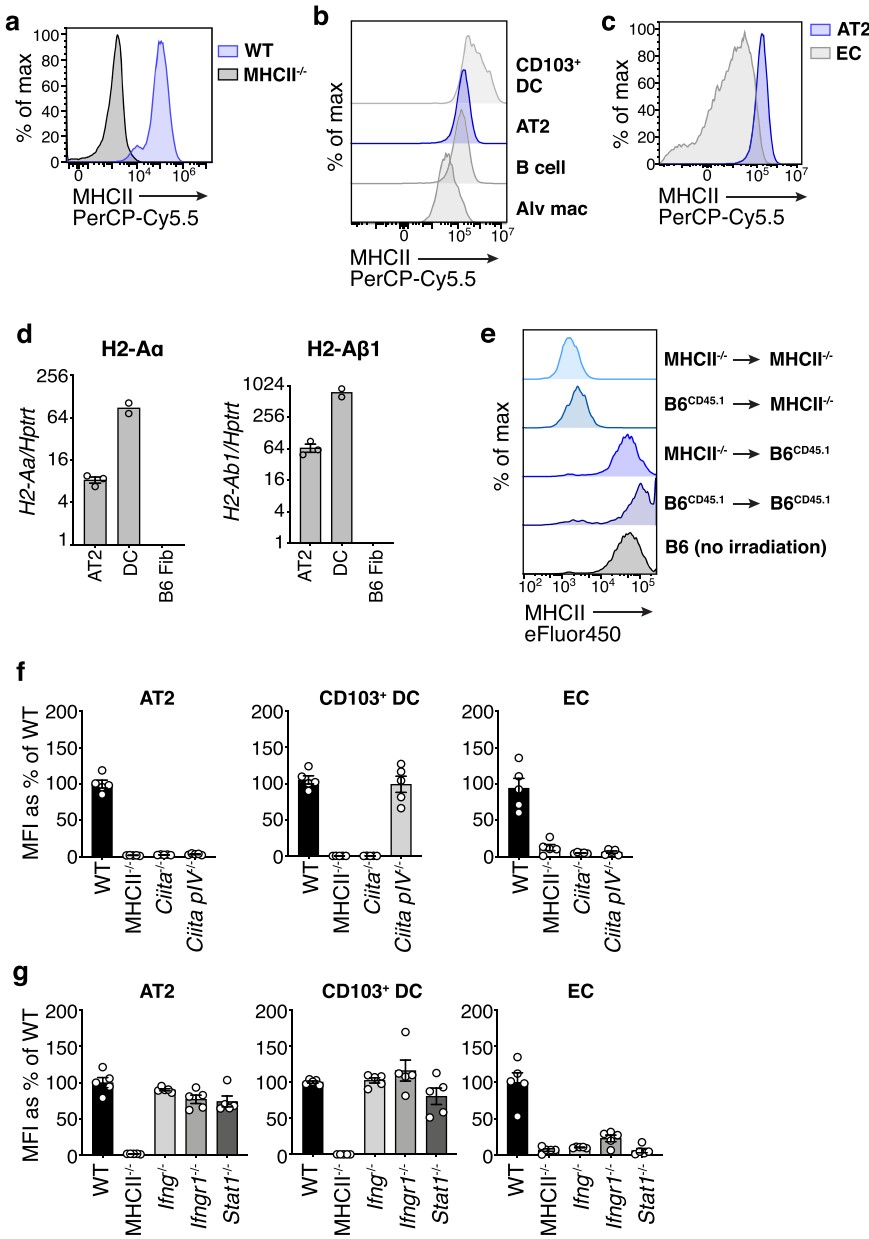

**Fig. 1 Type II alveolar cells (AT2s) constitutively express MHCII independent of conventional inflammatory signals. a–c** MHCII protein expression by AT2s from naive C57Bl/6 (B6) wild-type (WT) and MHCII$^{-/-}$ mice (**a**) and AT2s, CD103$^+$ dendritic cells (CD103$^+$ DC), B cells, alveolar macrophages (Alv mac), and endothelial cells (EC), from naive B6 WT lungs (**b, c**) measured by flow cytometry; histograms represent $n > 10$ mice per strain total from >5 experiments. WT AT2s are in blue (**a–c**), MHCII$^{-/-}$ AT2s in black (**a**), and other WT cell types in gray as labeled (**b, c**). **d** I-A$^b$ α (H2-Aα) and β (H2-Aβ1) chain transcript abundance relative to housekeeping gene *Hprt1*, in AT2s and lung dendritic cells (DC) sorted from naive B6 WT mice, and a B6 fibroblast cell line (B6 Fib), measured by quantitative RT-PCR; symbols are biological replicates, $n = 3$ mice for AT2s, $n = 2$ for DCs, $n = 1$ for the cell line, each calculated as the average of three technical replicates; one representative experiment shown of two similar experiments. **e** MHCII protein expression by AT2s in MHCII$^{-/-}$ and CD45.1 B6 (B6$^{CD45.1}$) bone marrow chimeras, measured 8 weeks post-transplantation, with different transfers shown in shades of blue as labeled; histograms represent $n = 2$ [B6$^{CD45.1}$ → MHCII$^{-/-}$], $n = 3$ [B6$^{CD45.1}$ → B6$^{CD45.1}$], $n = 3$ [MHCII$^{-/-}$ → MHCII$^{-/-}$], $n = 4$ [MHCII$^{-/-}$ → B6$^{CD45.1}$] mice. **f, g** MHCII protein expression by AT2s, CD103$^+$ DCs, ECs from naive B6 WT, MHCII$^{-/-}$ (**f, g**) *Ciita*$^{-/-}$, *Ciita* pIV$^{-/-}$ (**f**), *Ifng*$^{-/-}$, *Ifngr1*$^{-/-}$, and *Stat1*$^{-/-}$ (**g**) lungs, measured by flow cytometry. Each symbol reflects MHCII expression on the cell type indicated in $n = 1$ mouse, with $n = 5$ per strain total pooled from 2 experiments for all strains, except *Ifngr1*$^{-/-}$, which was pooled from three experiments. MHCII expression is quantified as a percentage reflecting the average median fluorescence intensity (MFI) of MHCII on each cell type in a given knockout mouse relative to the MHCII MFI on the same cell type in WT mice. 3/5 *Ciita* pIV$^{-/-}$ were 8 months old, 1/5 *Ifngr1*$^{-/-}$ were 5 months old. WT mice are in black, other knockout strains in shades of gray as labeled. **d, f, g** Bars are mean plus standard error of the mean (SEM). Source data are provided as a Source data file.

inducible lysosomal thiol reductase (GILT), and lysosomal/endosomal cathepsin proteases[44]. AT2s have been previously demonstrated to express cathepsins, in particular Cathepsin H (CtsH), which is critical for surfactant protein processing[45,46]. By RNA-sequencing analysis, we found that naive mouse AT2s

express transcripts for AEP, GILT, and a wide variety of cathepsin enzymes (Supplementary Fig. 5a). Furthermore, we directly assessed the activity of two cathepsins in AT2s that play major roles in antigen processing: Cathepsin D (CtsD) and Cathepsin L (CtsL). We found that AT2s exhibited readily detectable activity

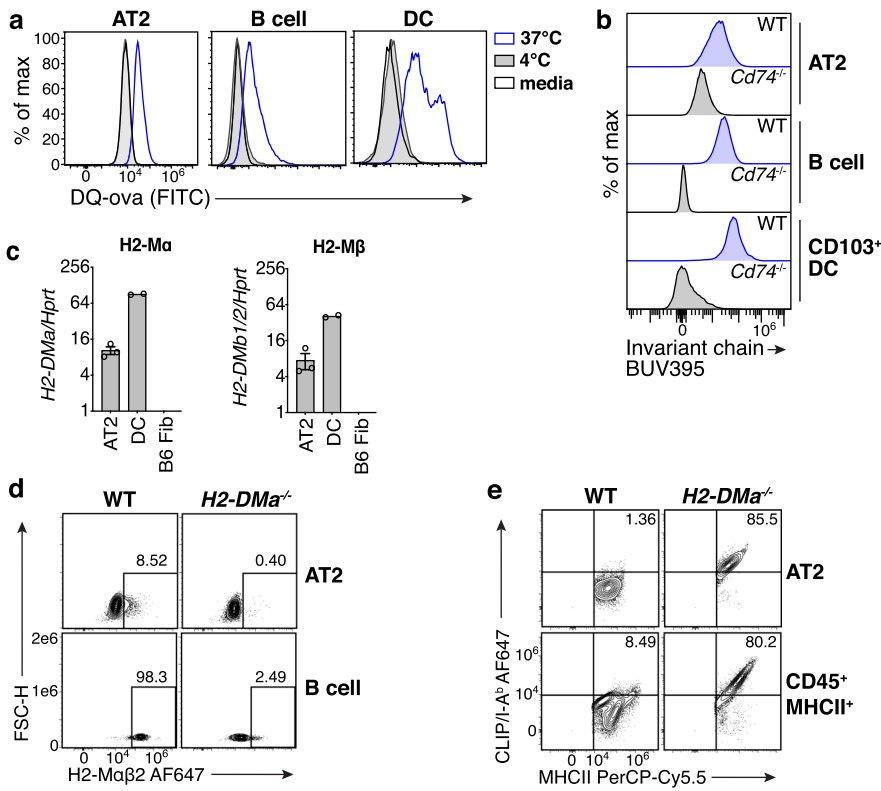

**Fig. 2 AT2s express canonical MHCII antigen processing and presentation machinery. a** Fluorescence of AT2s, B cells, and DCs sorted from naive B6 WT lungs then incubated with media only or DQ-ovalbumin (DQ-ova) at the temperatures indicated, measured by flow cytometry; histograms represent $n = 2$ mice total from two independent experiments. Media-only conditions are shown as black open histograms, 4 °C as gray filled, and 37 °C as blue open. **b** Intracellular invariant chain protein expression by AT2s, lung CD103[+] DCs, and splenic B cells from B6 WT (blue) or invariant chain deficient ($Cd74^{-/-}$, black) mice measured ex vivo by flow cytometry; histograms represent $n = 9$ WT and $n = 3$ $Cd74^{-/-}$ mice from three independent experiments. **c** H2-Mα and H2-Mβ chain transcript abundance relative to housekeeping gene $Hprt1$, in AT2s and DCs sorted from naive B6 WT lungs, and a B6 fibroblast cell line (B6 Fib), measured by quantitative RT-PCR; symbols are biological replicates, $n = 3$ mice for AT2s, $n = 2$ for DCs, $n = 1$ for the cell line, each calculated as the average of three technical replicates; one representative experiment shown of two similar experiments. **d** Intracellular H2-Mαβ2 protein expression by AT2s and lung B cells from B6 WT or H2-Mα deficient ($H2\text{-}DMa^{-/-}$) mice measured ex vivo by flow cytometry with frequency of H2-Mαβ2[+] cells shown; plots represent $n > 10$ mice per strain total from >5 independent experiments. Gates were drawn separately for AT2s and B cells based on their H2-Mα deficient cell counterparts. **e** Surface MHCII and CLIP/I-A[b] H2-Mαβ2 protein expression by AT2s and bulk CD45[+]MHCII[+] cells from B6 WT or H2-Mα deficient mice measured ex vivo by flow cytometry with frequency of MHCII[+]CLIP/I-A[b+] cells shown; plots represent $n > 6$ mice per strain total from >3 independent experiments. **c** Data are shown as mean plus SEM. Source data are provided as a Source data file.

of both CtsD and CtsL, which was abrogated by specific inhibitors of each enzyme (Supplementary Fig. 5b, c).

Optimal MHCII presentation also requires appropriate stabilization and trafficking of MHCII by the chaperone protein invariant chain[41–43]. We evaluated invariant chain expression in naive mouse AT2s by flow cytometry and, consistent with prior reports[22,38], found that AT2s express intracellular invariant chain (Fig. 2b).

Classical MHCII processing also depends on the protein H2-M, which catalyzes the removal of the invariant chain peptide remnant CLIP from the MHCII peptide-binding groove in the late endosome and facilitates the loading of higher affinity peptides[41–43]. We evaluated H2-M expression by qPCR, RNA-sequencing analysis, and flow cytometry. In mice, H2-M is a heterodimeric protein composed of one α chain pairing with one of two β chains, β1 or β2[47]; H2-Mαβ1 and H2-Mαβ2 are thought to function similarly[48,49]. By qPCR, AT2s expressed detectable levels of transcripts encoding the α and β chains of H2-M (Fig. 2c), using β primers that capture both β1 and β2. By flow cytometry AT2s did not express H2-Mαβ2 protein (Fig. 2d), suggesting that the predominant isoform of H2-M in naive mouse AT2s is therefore H2-Mαβ1, which was confirmed by RNA-

sequencing analysis (Supplementary Fig. 5a). We evaluated H2-M protein function in AT2s by flow cytometric detection of the CLIP/MHCII complex CLIP/I-A[b] in B6 mice. If H2-M is functionally present, then the number of CLIP/I-A[b] complexes will be low, as CLIP peptide will be edited out. AT2s from wild-type mice have low expression of CLIP/I-A[b] complexes, similar to hematopoietic MHCII[+] lung cells, and in contrast to both cell types from H2-M-deficient mice that have high levels of CLIP/I-A[b] (Fig. 2e). Furthermore, the peptide editing activity of H2-M can be antagonized by the protein H2-O[50]. We found that H2-O transcript and protein levels are undetectable in naive AT2s (Supplementary Fig. 5a, d). Altogether, these data demonstrate that AT2s express functional H2-M (H2-Mαβ1) protein.

We also assessed the expression of invariant chain protein and the human H2-M-equivalent, HLA-DM, in human AT2s, by flow cytometry. As in mice, healthy human AT2s expressed both the invariant chain and HLA-DM (Supplementary Fig. 2b).

Besides extracellular antigens, MHCII also presents peptides derived from intracellular proteins[51–53]. For example, in influenza (flu) virus infections in B6 mice, endogenous MHCII presentation of peptides from viral proteins synthesized within infected cells, instead of engulfed extracellular viral material, drives the majority

of the antiviral CD4+ T cell response[54]. Endogenous MHCII presentation involves a heterogenous network of machinery that is less well characterized compared to the classical pathway, precluding an exhaustive study of specific mediators here. However, one prerequisite for endogenous processing is expression of intracellular antigen by the APC. We assessed whether AT2s produce viral proteins intracellularly after exposure to live influenza virus and found that AT2s expressed both hemagglutinin (HA) as well as nucleoprotein (NP) to high levels (Supplementary Fig. 4). This suggests that AT2s possess abundant substrate material for endogenous processing during flu infections, and this presumably extends to other respiratory viral infections for which they are also main targets.

Beyond peptide/MHCII complex formation, another component of APC function is the provision of costimulation. In general, priming of naive T cells requires APC expression of the proteins CD80 or CD86, while the reactivation of effector T cells is less costimulation-dependent and may be influenced by a wider spectrum of costimulatory molecules[55,56]. Prior studies indicate that naive AT2s do not express CD80 or CD86, but that they express the noncanonical costimulatory molecule ICAM-1[19–21]. We also assessed AT2s for CD80, CD86, and ICAM-1 expression and found that AT2s expressed ICAM-1, but not CD80 or CD86, both at homeostasis and after influenza infection (Supplementary Fig. 5e, f).

Taken together, these data indicate that AT2s possess the requisite characteristics and intracellular machinery to endow them with MHCII antigen presentation capacity; in combination with the expression of ICAM-1 but not CD80/CD86, this would predict that AT2s have the ability to act as APCs, activating effector, but not naive, CD4+ T cells that enter the lung.

**AT2 MHCII contributes to improved respiratory viral disease outcomes.** As AT2s express both MHCII and the conventional MHCII-associated APC machinery, we predicted that AT2 MHCII presentation would contribute to immune responses in vivo. To test this, we generated mice with an AT2-specific deletion of MHCII by breeding mice to have both (1) a tamoxifen-inducible Cre enzyme controlled by the AT2-specific surfactant protein C (SPC) promoter[57] and (2) LoxP sites flanking exon 1 of the I-A^b β chain on both alleles[58] (SPC-Cre-ERT2^+/− × H2-Ab1^fl/fl; aka "SPC^ΔAb1"). Upon tamoxifen administration, SPC^ΔAb1 mice exhibit uniform deletion of MHCII from AT2s (Fig. 3a–c), in contrast to the parental strains whose AT2s retain MHCII: those expressing the Cre enzyme with a wild-type I-A^b locus (SPC-Cre-ERT2^+/− only; aka "SPC^Cre"), as well as those with homozygous floxed I-A^b alleles but no Cre (H2-Ab1^fl/fl only; aka "Ab1^fl/fl").

To ensure that the deletion of MHCII in SPC^ΔAb1 mice was restricted to AT2s, we compared MHCII on various other APCs in the lungs and spleens of all three strains. Similar frequencies of lung B cells, alveolar macrophages, lung CD103+ DCs, splenic B cells, and splenic CD8+ DCs, expressed MHCII in all three strains, suggesting that the Cre-mediated deletion of MHCII was AT2-specific (Fig. 3c). However, the overall magnitude of MHCII on certain APCs was lower in both the SPC^ΔAb1 and Ab1^fl/fl mice compared to the SPC^Cre strain (Fig. 3d); this suggests that in mice of the H2-Ab1^fl/fl genetic background, the I-A^b targeting construct diminishes MHCII expression in some cell types, independent of Cre expression. Based on these data, we considered the Ab1^fl/fl mice the most appropriate comparators for the SPC^ΔAb1 mice, so these two strains were paired for our in vivo studies.

We first assessed whether loss of MHCII on AT2s resulted in lung disease or immune dysfunction at homeostasis. To do so, we compared hematoxylin and eosin (H&E) staining of naive 12-week-old SPC^ΔAb1 and Ab1^fl/fl lungs. Both strains demonstrated normal, healthy lung architecture with minimal infiltrates (Fig. 3e). We also examined whether SPC^ΔAb1 mice had subclinical alterations in lung immune cells as well as splenic T cells, by flow cytometry. 12-week-old old naive SPC^ΔAb1 and Ab1^fl/fl lungs and spleens had overall similar frequencies and numbers of CD4+ and CD8+ T cells (Fig. 3f, Supplementary Fig. 6, Supplementary Table 1); both T cell subsets in both organs of the two strains also demonstrated similar expression of memory phenotype markers CD44 and CD62L (Fig. 3g, Supplementary Fig. 6), the proliferation marker Ki-67, and the inhibitory receptor and activation/exhaustion marker PD1 (Supplementary Fig. 6). Differences between SPC^ΔAb1 and Ab1^fl/fl mice did achieve statistical significance in the cases of CD4+ T cell numbers, the proportion of CD4s expressing CD44/CD62L, and the proportions of CD4s and CD8s expressing the activation/exhaustion marker PD1. However, these differences do not result in detectable pathology, as above (Fig. 3e), nor do they appear to influence outcomes following infection, as described below. As surface MHCII is the main ligand for the T cell inhibitory receptor LAG3, which restricts T cell expansion and effector function[59], we also evaluated LAG3-expressing T cells in the lungs and spleen of both strains at homeostasis; we found no differences in the frequencies of LAG3+ CD4+ and CD8+ T cells (Supplementary Fig. 7, Supplementary Table 2). The frequency of lung and splenic Tregs was also similar between SPC^ΔAb1 and Ab1^fl/fl mice (Fig. 3h, Supplementary Fig. 6). Frequencies of tissue resident-memory CD69+CD11a+ CD4+ and CD69+ CD8+ T cells[60] in the lungs were also similar between the two strains (Fig. 3i), as were the frequencies and numbers of alveolar macrophages, neutrophils, B cells, NK cells, and γδT cells (Supplementary Fig. 6). Thus, overall, MHCII appears to be dispensable for the maintenance of a healthy immune environment in the lung at homeostasis in young adult mice. Furthermore, as the SPC^ΔAb1 mice did not experience respiratory disease at homeostasis, and AT2s sorted from SPC^ΔAb1 and Ab1^fl/fl mice formed similar numbers of lung organoids in vitro (Supplementary Fig. 8), MHCII also does not seem to be required for two main physiologic functions of AT2s: surfactant production and lung regeneration.

We next assessed whether AT2 MHCII contributes to the outcome of lung infection. To do this, we used two respiratory virus infection models: mouse lung-adapted PR8 influenza A virus (IAV) as well as Sendai virus (SeV), a natural mouse pathogen similar to human parainfluenza virus. To assess the impact of AT2 MHCII on viral disease, we measured weight loss of SPC^ΔAb1 and Ab1^fl/fl mice after infection with either IAV or SeV. SPC^ΔAb1 and Ab1^fl/fl mice exhibited similar weight loss after IAV infection (Fig. 4a; Supplementary Table 3). However, after SeV infection, SPC^ΔAb1 mice experienced significantly more weight loss and delayed recovery compared to the Ab1^fl/fl group (Fig. 4b; Supplementary Table 4). We also assessed the impact of AT2 MHCII on mortality after infection with both viruses. SPC^ΔAb1 mice experienced reduced survival after IAV infection compared to Ab1^fl/fl controls, 24% vs 50%, respectively (Fig. 4c). Similarly, after SeV infection, SPC^ΔAb1 mice had worse survival with ~2-fold higher mortality (28%) compared to the Ab1^fl/fl controls (12%) (Fig. 4d). We next asked whether AT2 MHCII antigen presentation might contribute to the control of viral replication, by measuring lung virus titers after infection with either IAV or SeV. There were no significant differences in IAV titers 7 days after infection between SPC^ΔAb1 and Ab1^fl/fl mice (Fig. 4e). Similarly, SeV titers 4, 7, and 9 days after infection were similar between the two strains (Fig. 4f). As LAG3 has been shown to restrict lung CD4+ and CD8+ T cell responses during

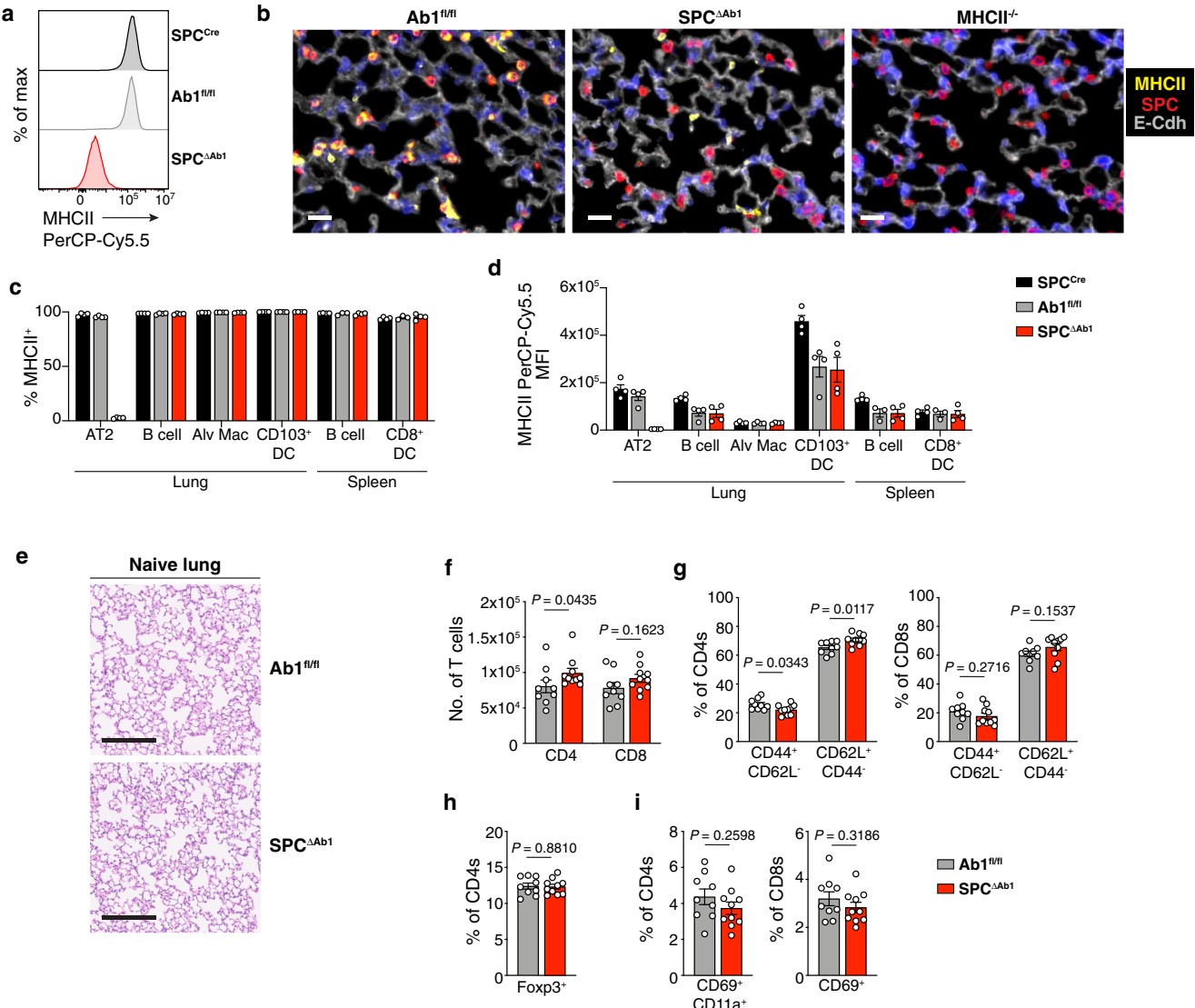

**Fig. 3 AT2 MHCII is dispensable for immune homeostasis in the lung in 12-week-old mice. a** MHCII expression by AT2s from naive *SPC-Cre-ERT2*[+/−] (SPC[Cre]), *H2-Ab1*[fl/fl] (Ab1[fl/fl]), and *SPC-Cre-ERT2*[+/−] × *H2-Ab1*[fl/fl] (SPC[ΔAb1]), lungs measured by flow cytometry; histograms represent *n* > 50 mice per strain total from >10 independent experiments. **b** Immunofluorescence detection of MHCII (yellow), SPC (red), E-Cadherin (gray), and DAPI (blue), in Ab1[fl/fl], SPC[ΔAb1], and MHCII[−/−] lungs; scale bar depicts 20 µm, representative images of *n* = 2 mice each for Ab1[fl/fl] and SPC[ΔAb1] strains and *n* = 1 for MHCII[−/−]. **c**, **d** Percent and MFI of each cell type expressing MHCII protein for lung AT2s, B cells, alveolar macrophages, CD103[+] DCs, and spleen B cells and CD8[+] DCs, from naive SPC[Cre], Ab1[fl/fl], and SPC[ΔAb1] mice, measured by ex vivo flow cytometry analysis; bars show *n* = 4 mice per strain from one experiment, representative of two similar independent experiments for the Ab1[fl/fl] and SPC[ΔAb1] strains (*n* = 8 mice total per strain). **e** H&E staining of naive 12-week-old Ab1[fl/fl] and SPC[ΔAb1] lungs, images representative of *n* = 5 mice per strain. Scale bar depicts 200 µm. **f–i** Absolute number of lung CD4[+] and CD8[+] T cells (**f**), and frequency of lung CD4[+] and CD8[+] T cells expressing CD44, CD62L (**g**), Foxp3 (**h**), and CD69, CD11a (**i**), as indicated, from naive 12-week-old Ab1[fl/fl] and SPC[ΔAb1] mice measured by ex vivo flow cytometry analysis; each symbol represents *n* = 1 mouse, with *n* = 9 Ab1[fl/fl] and *n* = 10 SPC[ΔAb1] mice total displayed from one experiment, which is representative of two similar experiments. **c**, **d**, **f–i** Data shown are mean plus SEM, analyzed by two-tailed Mann–Whitney test (**f** [CD4s]) or unpaired two-tailed Student's t-test (**f** [CD8s], **g–i**). Full statistical test results are in Supplementary Table 1. For **a**, **c**, **d**, **f–I** SPC[Cre] are in black, Ab1[fl/fl] in gray, SPC[ΔAb1] in red. Source data are provided as a Source data file.

respiratory viral infections[61,62], we also considered whether AT2 MHCII is required for LAG3-mediated T cell suppression during infection. We found similar frequencies and numbers of LAG3[+] CD4[+] and CD8[+] T cells in the lungs of SPC[ΔAb1] and Ab1[fl/fl] mice 9 days after IAV infection (Supplementary Fig. 7). Thus, while MHCII on AT2s is not required for protection against respiratory viral infection, it does contribute to lower morbidity and mortality, without impacting LAG3[+] T cell expansion or lung viral burden.

We also asked whether the statistically significant differences between certain lung T cell subsets in SPC[ΔAb1] and Ab1[fl/fl] mice

observed at homeostasis were amplified during infection. We observed no statistically significant differences in the numbers of lung CD4[+] T cells or the proportion of lung CD4s expressing CD44/CD62L between SPC[ΔAb1] and Ab1[fl/fl] mice 9 days after IAV infection (Supplementary Fig. 9a, b; Supplementary Table 5). Furthermore, the numbers and frequencies of lung CD4 and CD8 T cells expressing the activation/exhaustion marker PD1 were similar (Supplementary Fig. 9c, d), as was the proportion of AT2s expressing PD-L1, the cognate ligand for PD1 (Supplementary Fig. 9e). Thus, the differences in magnitude of these T cell subsets at homeostasis are unlikely to explain the differences in outcomes following infection.

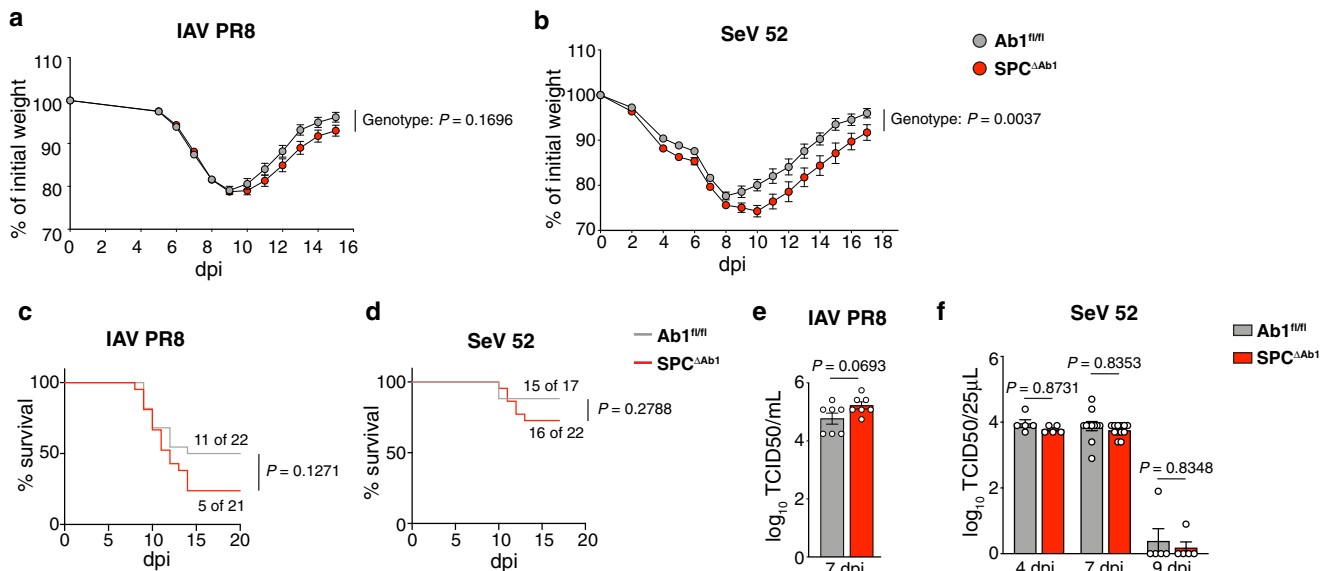

**Fig. 4 Loss of AT2 MHCII results in greater weight loss and reduced survival after respiratory viral infection. a–f** Comparison of Ab1$^{fl/fl}$ and SPC$^{\Delta Ab1}$ mice after infection with influenza strain PR8 (IAV PR8) (**a**, **c**, **e**) and Sendai virus strain 52 (SeV 52) (**b**, **d**, **f**). **a**, **b** Weight loss with weights displayed relative to day of infection; $n = 43$ Ab1$^{fl/fl}$ and $n = 53$ SPC$^{\Delta Ab1}$ mice pooled from six independent experiments (**a**), and $n = 17$ Ab1$^{fl/fl}$ and $n = 22$ SPC$^{\Delta Ab1}$ mice pooled from two independent experiments (**b**). **c**, **d** Mortality, with curves representing proportion surviving; $n = 22$ Ab1$^{fl/fl}$ and $n = 21$ SPC$^{\Delta Ab1}$ mice pooled from two independent experiments (**c**), and $n = 17$ Ab1$^{fl/fl}$ and $n = 22$ SPC$^{\Delta Ab1}$ mice pooled from two independent experiments (**d**). **e**, **f** Lung virus titers on the days post-infection (dpi) indicated; $n = 7$ mice per strain from one experiment (**e**), and $n = 5$ mice per strain for days 4 and 9, $n = 11$ Ab1$^{fl/fl}$ and $n = 12$ SPC$^{\Delta Ab1}$ mice for day 7, pooled from two independent experiments (**f**). **a**, **b** Data are mean plus SEM, analyzed by mixed-effects model with post-hoc multiple comparisons with Sidak's correction at each dpi. $P$ values displayed represent overall model effects. Full statistical test results are in Supplementary Tables 3–4. **c**, **d** Survival curves were compared via Log-rank test ($\chi^2 = 2.327$, df = 1 (**c**), and $\chi^2 = 1.173$, df = 1 (**d**)). **e**, **f** Bars are mean plus SEM of log$_{10}$ transformed values, analyzed by unpaired two-tailed Student's $t$-test ($t = 1.995$, df = 12) (**e**), and two-way ANOVA (genotype effect: $F = 1.559$, df.n = 1, df.d = 37, $P = 0.2197$) with post-hoc multiple comparisons with Sidak's correction (4 dpi $t = 0.6853$, df = 37; 7 dpi $t = 0.7603$, df = 37; 9 dpi $t = 0.7614$, df = 37). $P$ values displayed represent post-hoc comparisons (**f**). For all panels, Ab1$^{fl/fl}$ are in gray, SPC$^{\Delta Ab1}$ in red. Source data are provided as a Source data file.

In summary, these studies suggest that in vivo in young adult mice, AT2 MHCII expression is dispensable for healthy lung immune homeostasis but confers an appreciable advantage in respiratory viral disease outcome overall.

**AT2s exhibit restrained antigen presentation capacity via MHCII.** Although loss of AT2 MHCII worsened respiratory virus disease, the effect was smaller than anticipated based on the abundance of AT2s in the lung[32] and the magnitude of AT2 MHCII expression. A potential explanation for this more measured impact is that AT2s possess limited MHCII antigen processing and presentation capacity.

To investigate the antigen presentation function of AT2s, we first evaluated the ability of AT2s to stimulate flu peptide/MHCII complex-specific costimulation-independent T cell hybridomas (Fig. 5a). AT2 presentation of five different epitopes from live virus was undetectable (Fig. 5b, c), in contrast to professional APCs that presented all five. Poor presentation by AT2s was not due to a failure of in vitro infection, as AT2s were infected to levels higher than comparator professional APCs (Supplementary Fig. 4a). Limited presentation was observed across multiple MHCII alleles, B6 I-A$^b$ (Fig. 5b) and BALB/c I-E$^d$ (Fig. 5c), was not the result of protein source, as both neuraminidase (NA) and HA-derived epitopes were similarly affected (Fig. 5b, c), and was similar for epitopes generated by both exogenous (HA$_{107-119}$) and endogenous (NA$_{79-93}$) processing pathways[63] (Fig. 5c). Even when pulsed with synthetic peptides, AT2s were able to present only three of five epitopes (HA$_{91-107}$, HA$_{302-313}$, NA$_{79-93}$)

(Fig. 5b, c). Thus, AT2s exhibited a global impairment in the capacity to present MHCII-restricted epitopes in vitro.

Assays of AT2 function in vitro may be suboptimal due to reduced viability after cell sorting and the gradual de-differentiation of AT2s in standard tissue culture[64,65]. Additionally, measuring presentation on an individual epitope basis may underestimate the presentation of all possible MHCII-restricted flu peptides. Thus, we next examined AT2 in vivo flu peptide/MHCII complex formation, by co-culturing in vivo-infected AT2s from wild-type or MHCII$^{-/-}$ B6 mice taken 4 days post influenza infection with polyclonal splenic CD4$^+$ or CD8$^+$ T cells taken 9 days post-infection, in the presence of soluble anti-CD28 (Fig. 5d). MHCII presentation was detected via T cell IFNγ production captured by ELISpot. AT2s from flu-infected mice were capable of stimulating both CD4$^+$ and CD8$^+$ T cells to produce IFNγ (Fig. 5e). Stimulation of CD4$^+$ T cells, but not CD8$^+$ T cells, was abrogated when MHCII$^{-/-}$ flu-infected AT2s were used as APCs, confirming that stimulation of CD4$^+$ T cells by AT2s was MHCII-dependent. However, consistent with our in vitro studies, AT2 presentation to CD4$^+$ T cells was markedly less efficient than professional APCs, in this case CD103$^+$ CD11c$^+$ cells taken from the same lungs; despite being 9 times more infected, AT2s (99% infected, Supplementary Fig. 4b) stimulated less than half the IFNγ production than the comparator APCs (11% infected, Supplementary Fig. 4b).

We considered that the high degree of AT2 infection in our assays might impair their capacity to process and present antigen. To address this, we measured MHCII antigen presentation using a non-infectious model system, by staining lungs directly ex vivo

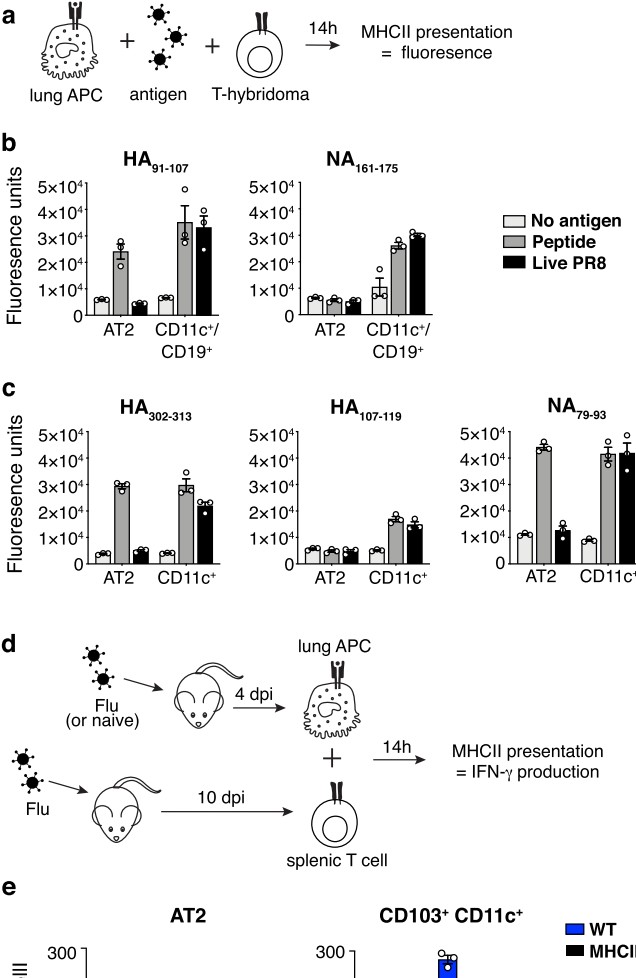

**Fig. 5 AT2s exhibit a globally restricted capacity to present influenza virus epitopes via MHCII. a** Hybridoma presentation assay. **b, c** Presentation of MHCII-restricted flu peptides by B6 AT2s and a mixed population of CD11c[+] and CD19[+] lung cells (**b**) or BALB/c AT2s and CD11c[+] lung cells (**c**) sorted from naive mice then incubated with synthetic peptide (dark gray), live virus (black), or no antigen (light gray), reflected by NFAT-*LacZ*-inducible T cell hybridoma activation and cleavage of a fluorogenic β-galactosidase substrate. Symbols shown represent technical replicates with $n = 3$ per condition, and bars reflect mean plus SEM. Data are from one experiment (**b**, **c**), which is representative of three similar independent experiments for (**b**). **d** Primary T cell ELISpot presentation assay. **e** Presentation of MHCII-restricted flu peptides by lung AT2s and CD103[+] CD11c[+] APCs sorted from naive or flu-infected B6 WT (blue) or MHCII[−/−] (black) mouse lungs 4 dpi, measured as the production of IFN-γ by responding flu-experienced splenic CD4[+] and CD8[+] T cells (as indicated); "APC %flu+" numbers describe the proportion of each APC type that was flu-infected (shown in Supplementary Fig. 4b). Symbols shown represent technical replicates from one experiment, representative of two similar independent experiments. Technical replicates displayed are: $n = 3$ for WT naive AT2 + CD4, and WT and MHCII[−/−] 4 dpi CD103[+] CD11c[+] + CD4 conditions; $n = 2$ for MHCII[−/−] naive AT2 + CD4, WT and MHCII[−/−] 4 dpi AT2 + CD4 conditions; $n = 1$ for WT and MHCII[−/−] 4dpi AT2 + CD8 conditions, WT and MHCII[−/−] naive CD103[+] CD11c[+] + CD4 conditions, and WT and MHCII[−/−] 4 dpi CD103[+] CD11c[+] + CD8 conditions. Bars represent averages of technical replicates for all conditions where $n = 2$–3, and single values where $n = 1$. Source data are provided as a Source data file.

AT2s after IFNγ administration (Fig. 6d, e), to levels significantly higher than that of B cells. IFNγ also induced some AT2s to express the H2-Mαβ2 isoform of H2-M (Fig. 6f, g), which is absent at baseline where AT2s express the H2-Mαβ1 isoform only, as previously demonstrated (Fig. 2, Supplementary Fig. 5a). Although H2-Mαβ2[+] AT2s exhibited more YAe staining than H2-Mαβ2[−] AT2s, both populations stained highly after IFNγ treatment (Fig. 6h), suggesting that a higher magnitude of H2-M contributes to enhanced Eα52-68/I-A[b] presentation, but the specific H2-Mαβ2 isoform is not required. Thus, AT2 MHCII presentation increases during IFNγ inflammation due to the upregulation of several MHCII presentation mediators, but it is still less efficient than that of professional APCs.

Taken together, these data demonstrate that AT2s are capable of presenting antigen via MHCII to CD4[+] T cells, and this is enhanced in the setting of inflammation; however, compared to professional APCs they exhibit limited antigen presentation of MHCII-restricted epitopes derived from both viral and endogenous proteins, as well as some extracellular peptides. This restrained presentation capacity may explain the variability seen in prior studies of AT2 function owing to differences in the model systems used, in particular the level of antigen provided to the AT2s, as well as the specific epitopes studied.

## Discussion

Here we establish that AT2 cells represent a unique category of MHCII-expressing cells, as non-immune cells that nonetheless constitutively express MHCII in an IFNγ-independent manner. We demonstrate that AT2 MHCII may contribute to improved outcomes of respiratory virus disease in vivo, and that in contrast to professional APCs, AT2s exhibit a program of restrained MHCII antigen presentation.

We were unable to identify an immune signal required for AT2 MHCII expression despite investigating a variety of broad adaptive and innate mediators, including the microbiota. This suggests that MHCII expression is not driven by inflammation. As AT2 MHCII is upregulated only after birth (J. Whitsett, B. Aronow, S. Potter

with the peptide/MHCII complex-specific "YAe" antibody[66] (Fig. 6a). YAe detects Eα52-68/I-A[b] complexes, which form in [BALB/c × B6] F1 mice as they are composed of a peptide from I-E[d] (BALB/c-derived) presented by I-A[b] (B6-derived). In naive F1 mice, AT2s had low yet detectable YAe staining above the background levels of B6 AT2s (Fig. 6b, c). However, YAe staining of F1 AT2s was markedly lower than that of F1 lung B cells (Fig. 6b, c). This could not be explained by differences in expression of the source proteins, I-A[b] and I-E[d], which are similarly expressed by AT2s and B cells at homeostasis, if not slightly higher in AT2s (Fig. 6d, e). These results indicate that AT2s are capable of forming Eα52-68/I-A[b] complexes at steady state, but they do so far less efficiently than do B cells.

To better approximate the setting of viral infection, we also treated mice with IFNγ and then measured Eα52-68/I-A[b] complex formation. AT2 YAe staining was substantially increased in F1 mice after treatment with IFNγ, but was again significantly lower than B cells in the same mice (Fig. 6b, c). To assess the contributors to increased MHCII presentation in AT2s, we evaluated changes in expression of I-A[b], I-E[d], and H2-M, all of which are required for Eα52-68/I-A[b] complex formation[67]. Both I-A[b] and I-E[d] increased in

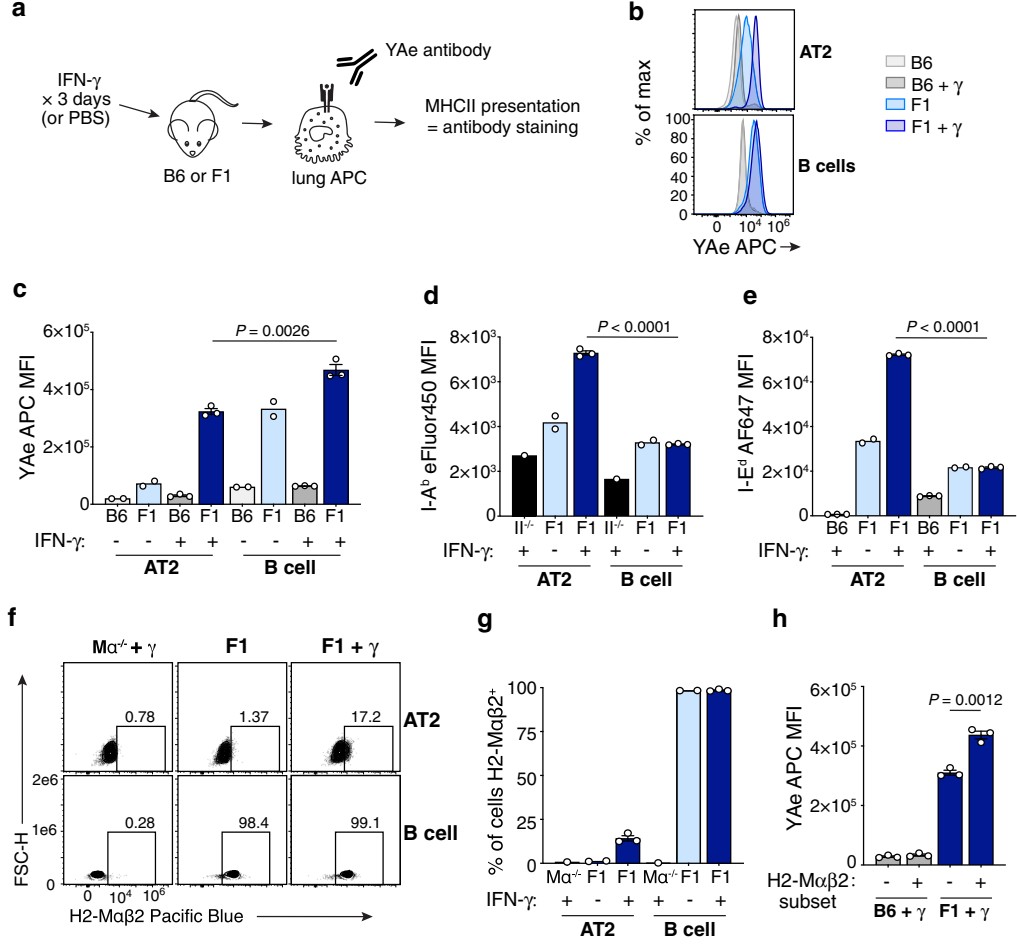

**Fig. 6 AT2 MHCII presentation is enhanced in the setting of inflammation but remains limited. a** YAe presentation assay. **b, c** Eα$_{52-68}$/I-A$^b$ complex formation by AT2s and B cells, detected by ex vivo YAe antibody staining of lungs from WT B6 or F1 [BALB/c × B6] mice treated with PBS or IFNγ. **d, e** I-A$^b$ (**d**) and I-E$^d$ (**e**) expression by AT2s and lung B cells, in MHCII$^{-/-}$ (II$^{-/-}$) B6 (**d**), WT B6 (**e**), or F1 (**d, e**) mice treated with PBS or IFNγ as indicated, detected by flow cytometry. **f, g** Intracellular H2-Mαβ2 expression by AT2s and lung B cells from F1 or $H2\text{-}DMa^{-/-}$ (Mα$^{-/-}$) B6 mice treated with PBS or IFNγ. Frequency of H2-Mαβ2$^+$ cells is shown above gates (**f**), which were drawn separately for AT2s and B cells based on the corresponding $H2\text{-}DMa^{-/-}$ cells. **h** YAe staining of H2-Mαβ2$^+$ and H2-Mαβ2$^-$ subpopulations of AT2s from B6 and F1 mice treated with IFNγ. **b, f** Plots represent at least $n = 4$ mice per group total from two independent experiments. **c-e, g, h** Each symbol represents $n = 1$ mouse. Displayed are $n = 2$ for conditions without IFNγ and $n = 3$ with IFNγ (**c**); $n = 1$ for MHCII$^{-/-}$, $n = 2$ for F1 without IFNγ, and $n = 3$ for F1 with IFNγ (**d**); $n = 2$ for F1 without IFNγ, and $n = 3$ for B6 and F1 with IFNγ (**e**); $n = 1$ for $H2\text{-}DMa^{-/-}$, $n = 2$ for F1 without IFNγ, and $n = 3$ for F1 with IFNγ (**g**); $n = 3$ for all (**h**). Data are from one experiment representative of two similar experiments (**c, g, h**) or from one experiment (**d, e**). Bars represent averages of biological replicates for conditions where $n = 2-3$ and single values where $n = 1$, with error bars as SEM. Data were analyzed by unpaired two-tailed Student's $t$-tests with [$t = 6.713$, df = 4] (**c**), [$t = 41.52$, df = 4] (**d**), [$t = 109.3$, df = 4] (**e**), and [$t = 8.165$, df = 4] (**h**). For **b-e, g, h**, as labeled, B6 + PBS conditions are in light gray, B6 + IFNγ in dark gray, F1 + PBS in light blue, F1 + IFNγ in dark blue, with MHCII$^{-/-}$ + IFNγ (**d**) and Mα$^{-/-}$ + IFNγ (**g**) in black. Source data are provided as a Source data file.

[CCHMC] and the LungMAP Consortium [U01HL122638]; single-cell RNA-seq data available at www.lungmap.net), its expression may instead be induced by a non-immunologic environmental stimulus or may be an intrinsic part of the AT2 cell developmental identity. The constitutive nature of AT2 MHCII expression prompted us to also consider whether AT2 MHCII plays a non-immune role in basic AT2 biology. We demonstrated that AT2 MHCII is not required for the regenerative function of AT2s, and as the SPC$^{\Delta Ab1}$ mice exhibit no lung disease either clinically or histologically, it is most likely also dispensable for surfactant production. More extensive characterization of AT2 physiology and function will be needed to investigate whether there are other non-immunologic processes that are disrupted by the loss of MHCII from AT2s.

We demonstrated that inefficient antigen presentation by AT2s cannot be explained by the absence of conventional MHCII-associated machinery. Instead, it is possible that there are

mechanisms operating in AT2s that actively oppose MHCII presentation. AT2s possess a highly specialized endosomal network specifically tailored to the synthesis, secretion, and recycling of surfactant proteins and lipids[46]. It is possible that these endocytic compartments are incompatible with the spatio-temporal degradative and peptide loading requirements for MHCII presentation. It is also unusual that AT2s present only some synthetic peptides when provided exogenously. One possible explanation is that AT2 cell surface MHCII molecules are already bound to a high affinity self-peptide that can be displaced only by even higher affinity peptides.

Restrained MHCII antigen presentation by AT2s is consistent with the more measured contribution they make via MHCII to the outcome of lung viral infection. We propose that this limited presentation is favorable in the lung, where excessive T cell activation would be especially damaging. In this model, high

MHCII expression poises AT2s to amplify lung adaptive immune responses, but restricted MHCII presentation establishes a higher threshold that must be overcome in order to do so. This system would allow for a more tempered response, where MHCII presentation by AT2s would be sufficient to trigger cognate T cell activation only in the setting of high antigen burden, such as a severe lung infection, but would prevent AT2-induced amplification of T cells in response to low levels of inhaled innocuous environmental antigens. Ongoing and future work will further explore this possibility.

We found that AT2 MHCII did not affect the total amount of virus in the lung, suggesting that AT2 MHCII contributes to protection by other mechanisms. AT2 MHCII may instead facilitate the development of a protective $CD4^+$ T cell cytokine milieu, which is critical for disease outcome in the setting of respiratory viral infection[68,69]. $CD4^+$ Tregs have also been demonstrated to facilitate lung repair by enhancing AT2 proliferation[70,71]. Thus, it is also possible that AT2 MHCII helps to facilitate disease recovery by promoting the expansion or localization of $CD4^+$ Tregs to the regions of the lung experiencing high virus loads, in order to minimize damage and enhance alveolar regeneration.

As lung-resident memory $CD4^+$ (Trm) can mediate protection upon viral challenge[72] and circulating virus-specific memory $CD4^+$ T cells correlate with heterosubtypic protection in humans[73], in future studies it will be important to address whether AT2 MHCII contributes to the development of virus-specific memory $CD4^+$ T cells. Prior studies from the Swain laboratory suggest that robust influenza-specific memory $CD4^+$ T cell formation requires two cognate TCR-MHCII encounters: the first priming interaction in the lymph node, and a second around days 5–7 post-infection[74,75]. As $MHCII^+$ AT2s are present at the site of infection with access to both antigen and antigen-specific $CD4^+$ T cells, it is possible that AT2s help to deliver this second checkpoint. It is also possible that local AT2 MHCII antigen presentation helps to facilitate the development and retention of lung-resident virus-specific memory $CD4^+$ Trm during infection[76–78].

AT2 MHCII may also play an important role in regulating $CD4^+$ T cell-driven lung diseases in response to respiratory allergens, such as in allergic asthma as well as hypersensitivity pneumonitis[79–82]. We demonstrated that the loss of AT2 MHCII did not result in lung disease at homeostasis in 12-week-old specific pathogen-free (SPF) mice, suggesting that AT2 MHCII does not contribute to inhaled antigen tolerance or allergy. However, as SPF mice experience artificially low levels of antigen exposure, it is possible that the impact of AT2 MHCII loss would be more apparent in mice with a greater burden or longer duration of inhaled putative allergen exposure, such as aged mice or wild mice[83–86], or those intentionally sensitized to allergens.

Finally, an essential avenue of future work will also be to understand whether alterations in AT2 MHCII expression or function contribute to the wide variation in outcome of infectious and immunologic lung diseases in humans. Of special interest at this time is whether AT2 MHCII contributes to protection in the setting of SARS-CoV-2 infection, as AT2s are the main cell type infected with this virus in the human lung parenchyma[87–89].

## Methods

**Mice**. C57Bl/6 wild-type (B6), B6.129S2-$H2^{dlAb1-Ea}$/J (MHCII$^{-/-}$)[90], B6;129S4-$H2$-$DMa^{tm1Luc}$/J ($H2$-$DMa^{-/-}$)[91], B6.129S7-$Ifng^{tm1Ts}$/J ($Ifng^{-/-}$)[92], B6.129S(Cg)-$Stat1^{tm1Dlv}$/J ($Stat1^{-/-}$)[93], C.129S2(B6)-$Ciita^{tm1Ccum}$/J ($Ciita^{-/-}$)[94,95], B6.SJL-$Ptprc^a$ $Pepc^b$/BoyJ (CD45.1 B6), B6.129X1-$H2$-$Ab1^{tm1Koni}$/J ($H2$-$Ab1^{fl/fl}$)[58], as well as BALB/c, and CB6F1/J (F1 [BALB/c × C57Bl/6] mice were originally purchased from the Jackson Laboratory. C57Bl/6 $Cd74^{-/-}$ mice[96] were originally provided by Guo-Ping Shi (Harvard), C57Bl/6 $Ciita$ $pIV^{-/-}$ mice[39] were originally provided by S. Hugues (University of Geneva), and C57Bl/6 $H2$-$Ob^{-/-}$ mice[97] were provided by L. Denzin (Rutgers). C57Bl/6 SPC-Cre-ERT2 mice[57] were originally provided by G.S. Worthen, B6.129S7-$Ifngr1^{tm1Agt}$/J ($Ifngr1^{-/-}$)[98], B6(Cg)-$Ifnar1^{tm1.2Ees}$/J ($Ifnar1^{-/-}$)[99], and B6.

Cg-$Ifngr1^{tm1Agt}$ $Ifnar1^{tm1.2Ees}$/J ($Ifnar1^{-/-}Ifngr1^{-/-}$) mice were provided by E. Behrens, and germ-free mice were provided by M. Silverman (Children's Hospital of Philadelphia). C57Bl/6 B6.129P2(SJL)-$Myd88^{tm1.1Defr}$/J ($Myd88^{-/-}$) mice[100] were provided by S. Shin, and B6.129S2(C)-$Stat6^{tm1Gru}$/J ($Stat6^{-/-}$) mice[101] were provided by C. Hunter (University of Pennsylvania). $Myd88^{-/-}$ and $Stat6^{-/-}$ mice were housed in specific pathogen-free facilities at the University of Pennsylvania, germ-free mice were housed in gnotobiotic mouse facilities at the University of Pennsylvania, and $H2$-$Ob^{-/-}$ mice were housed in the animal facility at Rutgers University. All other mice were maintained in specific pathogen-free facilities at the Children's Hospital of Philadelphia, at an ambient temperature of 68–74 °F with 30–70% humidity, with a 12 h light/dark cycle (lights on from 06:15 to 18:15, otherwise off). Mice were age and sex matched for all studies and 6–12-week-old mice were used for all experiments, unless otherwise specified. All animal procedures were in compliance with all relevant ethical guidelines for animal testing and research, including institutional and AAALAC guidelines. Animal protocols were approved by the Institutional Animal Care and Use Committee (IACUC) at the Children's Hospital of Philadelphia.

**Cell lines**. The B6 primary skin fibroblast cell line was derived in our laboratory and has been described previously[102]; it was maintained in Dulbecco's modified eagle medium (DMEM) containing 5% FBS, penicillin–streptomycin, and 2 mM L-glutamine. Madin-Darby canine kidney (MDCK) cells were provided by S. Hensley (University of Pennsylvania), originally obtained from the National Institutes of Health, and were maintained in MEM with 10% FBS. Monkey kidney LLC-MK2 cells (ATCC CCL-7) were provided by Carolina Lopez (University of Pennsylvania) and were maintained in DMEM with 10% heat-inactivated FBS, penicillin–streptomycin, 2 mM L-glutamine, and 1 mM sodium pyruvate. T cell hybridoma lines were derived in our laboratory and have been described previously[54,63]; these were maintained in "R10 media": RPMI media containing 10% FBS, 50 µM 2-mercaptoethanol, penicillin–streptomycin, 2 mM L-glutamine.

**Viruses**. Mouse lung-adapted H1N1 influenza A virus, A/Puerto Rico/8/1934 (IAV PR8), was originally provided by C. Lopez (University of Pennsylvania), and then expanded in 10-day-old embryonated chicken eggs. IAV PR8 viral titer was determined by standard focus-forming unit (FFU) assay in MDCK cells. Sendai virus, strain 52 (SeV 52), was also a generous gift of C. Lopez (University of Pennsylvania). SeV 52 viral titer was determined using a tissue culture infectious dose ($TCID_{50}$) standard infectivity assay in LLC-MK2 cells.

**Synthetic peptides**. The following synthetic peptides were used: $HA_{91-107}$ (RSWSYIVETPNSENGIC), $NA_{161-175}$ (SVAWSASACHDGMGW), $HA_{302-313}$ (CPKYVRSAKLRM), $HA_{107-119}$ (SVSSFERFEIFPK), $NA_{79-93}$ (IRGWAIYSKDN-SIRI). All peptides were obtained at >85% purity from Genscript.

**Tissue isolation for in vitro analysis and flow cytometry**. For isolation of AT2 cells, the lung vasculature was first perfused with 5 mL PBS by injecting into the cardiac right ventricle. Lungs were then inflated via intratracheal instillation of 0.9 mL AT2 digest media [5 U/mL Dispase (354235, Corning), 1.66 mg/mL Collagenase A (10103586001, Sigma), 0.33 mg/mL DNaseI (10104159001, Sigma) in PBS]. The trachea was then tied with suture to keep the lungs inflated while they were excised en bloc and then placed in an additional 1 mL digest media, then incubated at 37 °C. After 45 min, 5 mL 20% FBS in PBS was added, and the parenchymal lung lobes were removed from the large airways with forceps, then dissociated by vigorous pipetting. Digested lungs were passed through a 70 µm strainer, incubated in ACK lysis buffer to remove RBCs, then passed through a 40 µm strainer to obtain a single-cell suspension.

For isolation of lung T cells, the lung vasculature was first perfused with 5 mL 1% FBS in PBS by injecting into the cardiac right ventricle. Individual parenchymal lung lobes were removed from the chest cavity and placed into gentleMACS C tubes containing 2 mL 1% FBS in PBS. Lymphocyte digest media was then added to the lungs (1% FBS, 2.25 mg/mL Collagenase D (11088366001, Sigma), 0.15 mg/mL DNase I in PBS in 4 mL final volume), which were then gently disrupted using gentleMACS homogenizer program m_spleen_01.01, then incubated for 45 min at 37 °C with shaking. R10 media was then added to each tube, followed by further dissociation using gentleMACS homogenizer program m_lung_02.01. Digested lungs were then passed through a 70 µm strainer, incubated in ACK lysis buffer to remove RBCs, then passed through a 40 µm strainer to obtain a single-cell suspension.

AT2s could not be recovered from the T cell digest; likewise, the AT2 digest was not optimal for T cell isolation as it resulted in the degradation of surface CD4 and CD8 from T cells. In some T cell isolation experiments, a small sample of AT2s was also needed for the purposes of confirmation phenotyping to assess deletion or presence of MHCII. In these cases, prior to the addition of lymphocyte digest media to the lungs, a small ([<2 mm]$^3$) portion of lung was removed, and then incubated for 1 h at 37 C in 0.4 mL AT2 digest media. Next, 0.1 mL of FBS was added, and the sample was then homogenized through a 40 µm cell strainer using the flat end of a 1 mL syringe plunger, to obtain a single-cell suspension.

To harvest lungs for virus titering, lung lobes were removed directly from the thoracic cavity (with no perfusion or instillation) and placed in gentleMACS M

tubes containing 1 mL PBS. Additional PBS was then added to each M tube to produce a final 10% weight/volume solution of lungs/PBS. Lungs were then homogenized using gentleMACS dissociator program RNA_01.01. Cellular debris was removed by centrifugation at $600 \times g$ for 10" at 4 °C, and the clarified supernatant was then used for virus titering.

For isolation of splenocytes, spleens were removed from the abdominal cavity and placed directly in PBS. Spleens were then homogenized through a 70 μm cell strainer using the blunt end of a 3 mL syringe plunger. The splenocytes were then incubated in ACK lysis buffer to remove RBCs, then passed through a 40 μm strainer to obtain a single-cell suspension.

For isolation of immune cells from peripheral blood, blood was collected via cheek bleed or IVC puncture into PBS containing 25 mM EDTA. Samples were centrifuged at 300xg for 10", followed by ACK lysis of the cell pellet to remove RBCs.

For isolation of bone marrow cells, whole bones of the hindlimb were removed from mice. In a sterile manner, the ends of the bones were cut and the interior cavity was flushed with PBS.

For isolation of human distal lung cells, healthy human lungs were obtained with informed consent from the Prospective Registry of Outcomes in Patients Electing Lung Transplantation Study approved by University of Pennsylvania Institutional Review Board in accordance with institutional ethical procedures and guidelines. The donor used in this study had the following characteristics: 36 y.o. M, no smoking history, no evidence of active infection, P/F ratio of 346. Lungs were digested as described previously by Zacharias and Morrisey[103]. Briefly, a $2 \times 2$ cm piece of distal lung tissue (pleura and airways removed) was minced then processed in the same AT2 digest media as above using a gentleMACS dissociator at 37 °C for 35 min. Digested lungs were washed, passed through 70 μm and 40 μm strainers, and then incubated in ACK lysis buffer to remove RBCs and generate a single-cell suspension.

**Cell type identification via flow cytometry and cell sorting**. All flow cytometry antibodies and dilutions used are listed in Supplementary Table 6. For all flow cytometry experiments, unless otherwise stated, single-cell suspensions were first stained for dead-cell exclusion with Live/Dead Aqua (1:500, L34957, Thermo Fisher) in PBS for 15" at 4 °C, followed by Fc-receptor blockade with anti-CD16/CD32 (1:25, 553141, BD) for 5–10" at 4 °C for mouse cells or Human TruStain FcX (1:20, 422301, Biolegend) for 5–10" at RT for human cells, in 0.1% BSA in PBS. Cells were then stained with surface antibodies diluted 1:100 (unless otherwise specified) in 0.1% BSA in PBS for 30" in the dark at 4 °C. If necessary, cells were then fixed and permeabilized for intracellular staining using the BD Cytofix/Cytoperm kit (554714, BD) or for intranuclear staining using the eBioscience Foxp3/Transcription Factor Staining kit (00-5523-00, Thermo Fisher). Intracellular and intranuclear stains were diluted 1:100 (unless otherwise specified) in the corresponding kit wash buffer and were incubated for 30" at room temperature. Flow cytometry data were acquired on the following instruments: LSRII and LSRFortessa using FACSDiva Software (BD); CytoFLEX LX and CytoFLEX S using CytExpert software (Beckman Coulter). Cell sorting was performed on the following instruments: FACSAria Fusion using FACSDiva software (BD); FACSJazz using FACS Sortware software (BD); MoFlo Astrios using Summit software (Beckman Coulter). All flow cytometry analyses were conducted using FlowJo software (FlowJo LLC).

Murine AT2s were identified by flow cytometry and sorting as detailed in Supplementary Fig. 1[22]. In studies where MHCII expression was evaluated, AT2s were identified using the gating strategy without MHCII as a selection marker, as CD45⁻, CD31⁻, EpCAMint cells. For all other flow cytometry studies, AT2s were identified as CD45⁻, CD31⁻, EpCAM⁺ MHCII⁺ cells. Both gating strategies were validating by intracellular staining for pro-SPC followed by an anti-rabbit IgG fluorophore-conjugated secondary antibody.

Human AT2s were identified by flow cytometry as detailed in Supplementary Fig. 2 as HT2-280⁺ lung cells[35], by staining with unlabeled anti-HT2-280 (diluted 1:50) followed by an anti-mouse IgM fluorophore-conjugated secondary antibody.

Other cell types were identified via flow cytometry as follows: Lung endothelial cells (CD45⁻, CD31⁺), Lung CD103⁺ DCs (CD45⁺, CD11c⁺, CD103⁺), Lung alveolar macrophages (CD45⁺, CD11c⁺, CD11b⁻, CD64⁺), lung and splenic B cells (FSC-Alow, CD45⁺, CD11c⁻, CD19⁺), NK cells (CD45⁺, CD3⁻, NK1.1⁺), γδ T cells (CD45⁺, CD3⁺, CD11b⁻, TCRδ⁺), neutrophils (CD45⁺, CD11c⁻, CD11b⁺, Ly6G⁺), CD45⁺ MHCII⁺ cells (CD45⁺, MHCII⁺), splenic CD8⁺ DCs (CD45⁺, CD11chi, CD11b⁻, CD8⁺), CD4⁺ T cells (CD3⁺, CD4⁺, CD8⁻), and CD8⁺ T cells (CD3⁺, CD8⁺, CD4⁻). All gating strategies are demonstrated in Supplementary Figs. 10–11.

For cell sorting (FACS) experiments, samples were not stained with the Live/Dead viability dye or Fc-receptor blockade to minimize sample processing time and preserve cell viability. The following professional APC populations were sorted: qPCR DCs (CD45⁺, CD11c⁺, MHCIIhi); DQ-ova assay DCs (CD45⁺, CD11c⁺, MHCIIhi) and B cells (CD45⁺, B220⁺, MHCII⁺); Cathepsin L assay B cells (CD45⁺ CD19⁺); C57Bl/6 hybridoma assay CD11c⁺/CD19⁺ APCs (CD45⁺, MHCII⁺, CD11c⁺ or CD19⁺); Balb/c hybridoma assay CD11c⁺ APCs (CD45⁺, MHCII⁺, CD11c⁺); ELISpot CD103⁺ CD11c⁺ APCs (CD45⁺, CD11c⁺, CD103⁺); all are demonstrated in Supplementary Fig. 12. The following AT2 populations were sorted: DQ-ova assay, qPCR, C57Bl/6 hybridoma assay and Balb/c hybridoma

assay AT2s (CD45⁻, CD31⁻, EpCAM⁺, MHCII⁺), cathepsin D, cathepsin L, and ELISpot assay AT2s (CD45⁻, CD31⁻, EpCAM⁺); all are demonstrated in Supplementary Fig. 1. For organoid cultures, AT2 (CD45⁻, CD31⁻, Podoplanin⁻, CD34⁻, Sca1⁻, EpCAMint) and lung fibroblasts (Pdgfrα⁺; diluted 1:250) were sorted, as in Supplementary Fig. 12, and as demonstrated by Paris and colleagues[104].

**Flow cytometric detection of MHCII, peptide/MHCII complexes, and associated machinery**. To assess total surface MHCII expression in wild-type and knockout mice, cells were stained with a pan I-A/I-E anti-MHCII antibody. Human MHCII was detected using an anti-HLA-DR antibody. For detection of specific MHCII alleles in YAe presentation assays, cells were surface stained with anti-I-Ab and anti-I-Ed antibodies. Mouse invariant chain and H2-M expression were detected by staining intracellularly with anti-mouse CD74 and anti-H2-Mαβ2 antibodies, respectively. Two different fluorophore-conjugated versions of the anti-H2-Mαβ2 antibody were labeled in house, with the Pacific Blue and Alexa Fluor 647 labeling kits (P30013 or A20186, Thermo Fisher). Human invariant chain was detected using an anti-human CD74 antibody (diluted 1:500), and HLA-DM was detected using an anti-HLA-DM antibody[97] (diluted 1:500) provided by L. Denzin (Rutgers). CLIP-loaded MHCII molecules were detected on the cell surface by using an CLIP/I-Ab complex specific antibody[97] (diluted 1:500) provided by L. Denzin (Rutgers). To detect Eα₅₂₋₆₈/I-Ab complexes, lung cells were surface stained with a biotinylated Y-Ae antibody (diluted 1:50) followed by APC-Streptavidin (1:500, 405207, Biolegend). Mouse H2-O expression was detected by staining intracellularly with an anti-mouse H2-Oβ antibody[97] (diluted 1:500) provided by L. Denzin (Rutgers).

**RNA isolation and quantitative PCR (qPCR)**. Total RNA was extracted and purified using the Qiaqen RNeasy Plus Mini Kit (74134, Qiagen), and cDNA was then prepared using the SuperScript III First-Strand Synthesis System with random hexamers (18080051, Thermo Fisher). Quantitative PCR was performed using the Power SYBR Green PCR Master Mix system (4367659, Thermo Fisher) measured using a StepOnePlus Real Time PCR Machine with StepOnePlus software (Applied Biosystems). Expression was quantified relative to the housekeeping gene Hprt using the $\Delta C_T$ method. The following forward (F) and reverse (R) primer pairs were used and are also listed in Supplementary Table 7. H2-Aa F: CTGATTCT GGGGGTCCTCGC and R: CCTACGTGGTCGGCCTCAAT; H2-Ab1 F: GAGC AAGATGTTGAGCGGCA and R: GCCTCGAGGTCCTTTCTGACTC; H2-DMa F: GGCGGTGCTCGAAGCA and R: TGTGCCGGAATGTGTGGTT; H2-DMb1 F: CTATCCAGCGGATGTGACCAT and R: TGGGCTGAGCCGTCTTCT; Hprt1 F: TCAGTCAACGGGGGACATAA and R: GGGGCTGTACTGCTTAACCAG.

**Bone marrow chimeras**. Donor bone marrow was isolated from B6 MHCII⁻/⁻ and CD45.1 B6 mice, and T cells were depleted magnetically using Thy1.2 Dynabeads (11443D, Thermo Fisher). $5 \times 10^6$ bone marrow cells were then injected intravenously into lethally irradiated recipient B6 MHCII⁻/⁻ and CD45.1 B6 mice 6 h after irradiation was completed ($5.5$ Gy × 2 doses, 3 h apart). Mice were treated with Bactrim for 3 weeks following the transfer and were housed in autoclaved cages.

Full bone marrow chimera reconstitution was confirmed 6 weeks after transfer, by flow cytometry analysis of peripheral blood cells stained with anti-CD45.1 and anti-CD45.2 antibodies, as demonstrated in Supplementary Fig. 13. MHCII expression on chimeric mouse lung cells was assessed 8 weeks post-transfer.

**DQ-ovalbumin assay**. $0.75–1.25 \times 10^5$ AT2s, DCs, and B cells were incubated with either 0 or 10 μg/mL DQ-ovalbumin (D12053, Thermo Fisher) in 10% FBS in RPMI media at either 37 °C or 4 °C for 2 h. Cells were then washed 3× with PBS and stained with Live/Dead Aqua; cells were maintained at 4 °C during these steps. Fluorescence was then immediately assessed by flow cytometry.

**RNA-sequencing analysis**. FASTQ files were obtained using the NCBI SRA toolkit from the GEO accession GSE115904, originally provided by Ma et al.[38], DOI: 10.1128/JVI.01986-18. Salmon quasi-alignment was then used to normalize and quantify gene expression to generate a transcripts per million (tpm) matrix[105]. This was performed using the "validateMappings" setting and the pre-computed partial mm10 index for Salmon provided on refgenie. The generated matrices were then imported into R v4.0.2 for conversion and concatenation of Ensembl transcript version IDs to gene symbols using the biomaRt package[106].

**Cathepsin D and L activity assays**. Cathepsin D (CtsD) activity was measured using the SensoLyte ® 520 Cathepsin D Assay Kit (AS-72170, Anaspec) per the manufacturer's protocol, with lysates from $1 \times 10^5$ cells plated in each well. Cathepsin L (CtsL) activity was measured using the SensoLyte ® Rh110 Cathepsin L Assay Kit (AS-72217, Anaspec) per the manufacturer's protocol, with lysates from $4 \times 10^5$ cells plated in each well.

For both assays, splenocytes were harvested as above, and AT2s were isolated via FACS, as above. For the CtsD activity assay, splenic B cells were isolated from splenocytes using the DynabeadsTM Mouse CD43 UntouchedTM B Cells kit

(11422D, Thermo Fisher), per the manufacturer's protocol. For the CtsL activity assay, lung B cells were isolated via FACS, as above.

**Costimulatory molecule expression analysis.** To generate bone marrow-derived dendritic cells (BMDCs), bone marrow cells were grown in R10 media supplemented with 20 ng/mL mouse recombinant granulocyte-macrophage colony-stimulating factor ("GM-CSF"; 10822-026, Shenandoah Biotechnology). Fresh media was added on days 3 and 6 after plating, and the floating fraction of cells was harvested on day 7–10.

Mice were infected intranasally with 60 FFU IAV PR8 diluted in 20 μL PBS under isoflurane anesthesia. Lungs were harvested from mice 5 days after infection or from naive mice.

To evaluate the expression of costimulatory molecules, BMDCs, bulk splenocytes, and lung cells were surface stained with anti-CD80, anti-CD86, and anti-CD54 (ICAM1) antibodies (all diluted 1:50).

**Detection of germline MHCII deletion.** Consistent with prior reports[107,108], our Ab1$^{fl/fl}$ mice experienced spontaneous germline disruption of the I-A$^b$ locus at a rate of ~5%, resulting in global loss of MHCII from all cells. Therefore, in addition to standard genotyping all SPC$^{\Delta\Delta b1}$ and Ab1$^{fl/fl}$ mice were screened at >5 weeks of age for germline deletion of MHCII, by flow cytometric measurement of MHCII on peripheral blood immune cells isolated by cheek bleed. Specifically, cells were stained for surface expression of CD45, CD19, and MHCII, as demonstrated in Supplementary Fig. 13, and mice were excluded from further study if B cells (CD45$^+$ CD19$^+$) lacked MHCII expression. All mice included in experiments discussed here had appropriately intact MHCII expression by peripheral blood screening.

**Tamoxifen administration.** SPC$^{\Delta\Delta b1}$, Ab1$^{fl/fl}$, and SPC$^{Cre}$ mice >5 weeks of age were oral gavaged daily with 2 mg tamoxifen for 4 days, by delivering 100 μL of 20 mg/mL solution of tamoxifen (T5648, Sigma) in a 9:1 corn oil:ethanol mixture. Mice were used in experiments ≥5 days after the last dose of tamoxifen.

**Histological analysis.** To assess MHCII expression in SPC$^{\Delta\Delta b1}$, Ab1$^{fl/fl}$, and MHCII$^{-/-}$ mice by immunofluorescence, the lung vasculature was first perfused with 5 mL 4% paraformaldehyde in PBS (4% PFA) by injecting into the cardiac right ventricle. Lungs were then inflated via intratracheal instillation of 0.9 mL 4% PFA. The trachea was then tied with suture to keep the lungs inflated while they were excised en bloc and then placed in an additional 30 mL 4% PFA, then incubated for 24 h at 4 °C. Tissue was then processed, paraffin-embedded, sectioned, and stained with DAPI, and the following antibodies: anti-mouse pro-SPC (AB3786, EMD Millipore), anti-mouse MHCII (107601, Biolegend), and anti-mouse E-cadherin (ab76319, Abcam). Images were acquired using an Axio Observer 7 widefield microscope with Axiocam 702 monochrome CMOS camera and Zen blue acquisition software (Zeiss). Composite images were then generated in Fiji.

To assess lung pathology at homeostasis in SPC$^{\Delta\Delta b1}$ and Ab1$^{fl/fl}$ mice, the lungs were inflated via intratracheal instillation of ~1 mL 10% neutral buffered formalin (NBF), then placed in an additional ~30 mL 10% NBF and incubated for 24 h at RT. Tissue was then processed, paraffin-embedded, sectioned, and stained with hematoxylin and eosin, and the resulting lung sections were evaluated for signs of disease by a veterinary pathologist. Images were acquired using an Aperio AT2 Slide Scanner (Leica) and evaluated using Aperio ImageScope software (Leica).

**Homeostasis T cell phenotyping analyses.** For determination of T cell phenotype as well as activation status, cells were surface stained for the following markers: CD44, CD62L, CD69, CD11a, PD1, and LAG3. Cells were also stained intranuclearly for FoxP3 and Ki67. All gating strategies are demonstrated in Supplementary Fig. 11.

**Lung organoids.** Alveolar organoids were cultured as described previously[29,31], with the following modifications. Briefly, $5 \times 10^3$ AT2s sorted from either SPC$^{\Delta\Delta b1}$ and Ab1$^{fl/fl}$ mice were co-cultured with $5 \times 10^4$ Pdgfrα$^+$ lung fibroblasts sorted from WT C57Bl/6 mice. Cells were suspended in 90μL of a 1:1 mixture of Matrigel: modified SAGM media (CC-3118, Lonza). SAGM media is modified by adding 0.1 μg/mL cholera toxin (C8052, Sigma) and the following SAGM BulletKit components: 10 μg/mL insulin, 5 μg/mL transferrin, 25 ng/mL EGF, 30 μg/mL bovine pituitary extract, 0.01 μM retinoic acid, and 5% FBS. The 1:1 mixture containing cells was placed in a cell culture insert (353095, Corning); after solidification of the matrix at 37 °C, the insert was placed inside a well of 24-well plate containing modified SAGM media. For the first two days of culture, 10 μM ROCK inhibitor Y27632 (Y0503, Sigma) was added to the media. The media was changed every 48 h, and after 21 days organoids were imaged using an EVOS FL Auto using built-in EVOS FL Auto software. Organoid numbers were quantified in Fiji using the Cell Counter plugin.

**Virus infections for weight loss and mortality analysis.** For influenza infection weight loss monitoring, mice were infected intranasally with 3 FFU IAV PR8 diluted in 20 μL PBS under isoflurane anesthesia. For influenza infection mortality

experiments, mice were infected intranasally with 15 FFU IAV PR8 diluted in 20 μL PBS under isoflurane anesthesia. For both Sendai virus infection weight loss monitoring and mortality analyses, mice were infected intranasally with $3.5–7 \times 10^4$ TCID$_{50}$ SeV 52 diluted in 35 μL PBS under ketamine (70 mg/kg) + xylazine (5 mg/kg) anesthesia. For all weight loss experiments, mice were weighed on the days indicated. For all mortality studies, mice were monitored daily for death or moribund state as an endpoint.

**Lung virus titering.** For influenza virus titering, mice were infected intranasally with 3 FFU IAV PR8 diluted in 35 μL PBS under ketamine (70 mg/kg) + xylazine (5 mg/kg) anesthesia. For Sendai virus titering, mice were infected intranasally with $7 \times 10^4$ TCID$_{50}$ SeV 52 diluted in 35 μL PBS under ketamine (70 mg/kg) + xylazine (5 mg/kg) anesthesia. Mice were sacrificed at the timepoints indicated for virus titers analysis.

For influenza virus titers determination, MDCK cells were infected with serial 10-fold dilutions of lung homogenates in MEM containing 50 μg/mL gentamicin, 5 mM HEPES, and 1 μg/mL TPCK-treated trypsin (LS003750, Worthington Biochemical) in quadruplicate per lung. After 4 days of incubation at 37 °C, the presence of virus at each dilution was determined by visual cytopathic effect, and lung virus titers were determined using the Reed and Muench Calculation.

For Sendai virus titers determination, LLC-MK2 cells were infected with serial 10-fold dilutions of lung homogenates in DMEM containing 50 μg/mL gentamicin, 0.35% BSA, 0.12% NaHCO$_3$, and 2 μg/mL TPCK-treated trypsin in triplicate per lung. After 3 days of incubation at 37 °C, the presence of virus was determined by assessing the capacity of 25 μL of the supernatant at each dilution to hemagglutinate 0.25% chicken red blood cells (cRBCs) in a 100 μL final volume after a 30" incubation at room temperature. Lung virus titers were then determined using the Reed and Muench Calculation.

**Influenza LAG3$^+$ T cell expansion analysis.** Mice were infected intranasally with 3 FFU IAV PR8 diluted in 20 μL PBS under isoflurane anesthesia. Mice were sacrificed 9 days after infection and lungs were stained ex vivo for T cell surface markers and LAG3, as demonstrated in Supplementary Fig. 11.

**Influenza lung PD-L1 and CD44, CD62L, and PD1 expression analysis.** Mice were infected intranasally with 3 FFU IAV PR8 diluted in 20 μL PBS under isoflurane anesthesia. For PD-L1 analysis, mice were sacrificed 9 days after infection and lungs were dissociated for AT2s as above, then stained ex vivo for AT2 cell markers and PD-L1, as demonstrated in Supplementary Fig. 11. For PD1, CD44, CD62L analysis, mice were sacrificed 9 days after infection and lungs were dissociated for T cells as above, then stained ex vivo for T cell surface markers, PD1, CD44, and CD62L, as demonstrated in Supplementary Fig. 11.

**Hybridoma assay.** NFAT-$lacZ$-inducible T cell hybridomas recognizing MHCII-restricted influenza-derived epitopes have been described previously[54,63]. Hybridoma recognition of cognate peptide/MHCII complexes results in β-galactosidase production, which was detected using a fluorometric β-galactosidase substrate 4-methyl-umbelliferyl-β-D-galactopyranoside ("MUG"; M1633, Sigma).

$1 \times 10^4$ APCs were treated with media only, 20 μg/mL peptide, or $1 \times 10^6$ FFU influenza virus in R10 media for 45" at 37 °C, 6% CO$_2$ in a 384 well plate; for AT2 conditions, wells were pre-coated with a thin layer of 9:1 R10:Matrigel (356231, Corning) mixture. After 45", $2 \times 10^4$ hybridomas were directly added, and cells were then co-cultured for 14–18 h at 37 °C, 6% CO$_2$. MUG substrate solution (33 μg/mL MUG in PBS containing 38.5 μM 2-mercaptoethanol, 9 mM MgCl2, and 1.25% NP40) was then added to the co-culture at a ratio of 1:5 MUG solution:co-culture, then incubated for 3 h at 37 °C, 6% CO$_2$. Fluorescence (excitation: 365 nm, emission: 445 nm) was detected using an Infinite M200 Pro Plate Reader with i-control software (Tecan).

**IFNγ ELISpot assay.** For isolation of influenza-infected APCs, WT and MHCII$^{-/-}$ mice were infected intranasally with $1.2 \times 10^6$ FFU IAV PR8 diluted in 35 μL PBS under ketamine (70 mg/kg) + xylazine (5 mg/kg) anesthesia. Lungs were harvested from mice 4 days after infection or from naive mice. APCs were isolated via cell sorting, as above.

For isolation of influenza-experienced effector T cells, WT mice were infected intranasally with 3 FFU IAV PR8 diluted in 35 μL PBS under ketamine (70 mg/kg) + xylazine (5 mg/kg) anesthesia. Spleens were harvested from mice 10 days after infection. CD4$^+$ and CD8$^+$ T cells were isolated from bulk splenocytes using Dynabeads Untouched Kits for mouse CD4 cells and CD8 cells, respectively, per the manufacturer's protocol (11415D and 11417D, Thermo Fisher).

96-well MultiScreen$_{HTS}$ IP 0.45 μm filter plates (MSIPS4W10, Millipore-Sigma) were coated with mouse IFNγ capture antibody (551881, BD) and incubated at 4 °C for 20 h prior to the assay. Plates were blocked with R10 media for >1 h at 37 °C, after which $5 \times 10^4$ APCs and $1 \times 10^5$ T cells were co-cultured for 14–18 h in the presence of 2 μg/mL anti-CD28 (40-0281-M001, Tonbo Biosciences). T cell IFNγ production was detected via biotinylated IFNγ detection antibody (551881, BD), followed by HRP-Streptavidin (557630, BD) and chromogenic 3-amino-9-ethylcarbazole (AEC) substrate (551951). IFNγ spots were imaged and counted using a CTL ImmunoSpot S6 Universal Analyzer with ImmunoSpot software (ImmunoSpot).

**Detection of influenza-infected cells**. For quantification of influenza virus infection in hybridoma and ELISpot assays, at assay endpoint cells were stained with antibodies against surface hemagglutinin (HA) protein (diluted 1:50) or intracellular nucleoprotein (NP) protein. Two different fluorophore-conjugated versions of the anti-HA antibody were labeled in house, with the Alexa Fluor 488 and Alexa Fluor 647 labeling kits (A20181 or A20186, Thermo Fisher).

**IFNγ treatment of mice**. Mice were injected intravenously under isoflurane anesthesia once daily with $1 \times 10^5$ U recombinant mouse IFNγ (575306, Biolegend) in 150 μL PBS, or PBS only, for 3 days in a row.

**Figures and assay schematics**. All final figures were composed for publication in Adobe Illustrator. Assay diagrams in Figs. 5a, d, and 6a are original and were created for this manuscript using Adobe Illustrator. They are modeled in part after cartoon images originally from BioRender.com, which was accessed with a paid subscription that includes permission for journal publishing.

**Statistical analysis**. General data documentation was performed in Microsoft Excel. All statistical analyses were performed using Prism 8 software (GraphPad). The statistical tests used are indicated in the figure legend corresponding to each specific experiment. Unpaired two-tailed Student's t-test was used to compare means between two groups if they had similar variances by F-test, and if they passed all four of the following normality of residuals tests: Anderson-Darling, D'Agostino-Pearson omnibus, Shapiro–Wilk, and Kolmogorov–Smirnov. If variances were significantly different, unpaired two-tailed Welch's t-test was used, and if the samples failed at least one normality test then two-tailed Mann–Whitney test was used. For weight loss analysis, a mixed model, which uses a compound symmetry covariance matrix and is fit using restricted maximum likelihood (REML), with Geisser–Greenhouse correction was used, followed by post-hoc multiple comparisons with Sidak's correction; this was used instead of repeated-measures ANOVA, as it can handle missing values. For mortality analysis, the Log-rank (Mantel-cox) test was used to compare survival curves. Two-way ANOVA followed by post-hoc multiple comparisons with Sidak's correction was used to compare means between groups with 2 contributing independent variables.

**Reporting summary**. Further information on research design is available in the Nature Research Reporting Summary linked to this article.

## Data availability

All original data generated in this study are either available within the paper and its supplementary information files or are available from the corresponding author upon reasonable request. The FASTQ files used in our RNA-sequencing analysis were originally generated by Ma et al.,[38] they are publicly available via the GEO accession viewer as GSE 115904 and can be found at the following URL: https://www.ncbi.nlm.nih.gov/geo/query/acc.cgi?acc=GSE115904. Source data are provided with this paper.

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

## Acknowledgements

We thank the Children's Hospital of Philadelphia (CHOP) Flow Cytometry Core, in particular F. Tuluc, for their assistance with the flow cytometry and cell sorting experiments. We thank E. Radaelli and the Penn Vet Pathology Core for their help with our tissue histology studies, and we thank A. Stout and the Penn CBD Microscopy Core for their help with imaging. We thank the CHOP Abramson Research Center Animal Facility staff for their assistance with mouse colony maintenance. We thank C. Lopez and D. Fisher (University of Pennsylvania) for their assistance with the Sendai virus work and L. Denzin (Rutgers) for providing reagents and helpful discussion. We thank J. Ma and colleagues (University of Melbourne) for depositing their RNA-sequencing FASTQ files on GEO. The work was supported by National Institutes of Health grants 5R01AI113286 (L.C.E), T32AI007324 (S.A.T.), F30HL145907 (S.A.T), and the CHOP Division of Protective Immunity and Immunopathology Pilot Grant (S.A.T).

## Author contributions

S.A.T. and L.C.E. conceptualized the project. S.A.T. designed and performed all of the experiments, except for the RNA-seq analysis (Supplementary Fig. 5a), which was conducted by J.H.L. and the growth of organoids (Supplementary Fig. 8), which was conducted by A.J.P. L.C.E. supervised and provided guidance for all studies. C.B. provided critical assistance with the in vivo mouse studies. A.J.P. and G.S.W. contributed essential guidance for the study and isolation of AT2s and provided the SPC-Cre-ERT2 mice. J.K., M.C.B., and E.E.M. established the pipeline through which human AT2s were acquired (Supplementary Fig. 2). S.A.T. analyzed the data, made the figures, and wrote the original draft. L.C.E. edited the manuscript and figures. All authors reviewed the manuscript.

## Competing interests

The authors declare no competing interests.
