## [Peer Review File · Nature Communications]

REVIEWER COMMENTS

Reviewer #1 (Remarks to the Author):

Toulmin et al. comprehensively characterize the curious expression of MHC class II molecules (MHC II) in type II alveolar cells (AT2). This is a particularly timely study, given the high expression of ACE2, the SARS-CoV-2 receptor, on A2T cells, and their critical role in COVID 19.

The constitutive expression of MHC II on non-professional APCs is an oddity, and it has been known for some time that MHC II is expressed on AT2. Until this paper, however, the function of these molecules in antigen presentation and protection against lung infections was completely uncharacterized.

In this marvelous study, the authors show that AT2 MHC II expression is regulated in a manner that is independent of IFN- γ , and indeed, even CIITA, the master regulator of MHC II. This finding alone would make the paper worthy of publication in Nat Comm, but the authors go on to provide the first functional characterization of AT2 antigen presentation. They demonstrate that AT2 cells express both invariant chain and functional H2M, which appropriately reduces MHC II CLIP peptide expression.

By selectively knocking out MHC II in AT2 cells in vivo, Toulmin et al. establish that MHC II expression is not required for AT2 function or maintaining baseline lung immunity, but are required for optimal recovery from IAV or Sendai virus infection. This sets the stage for their most important, if mysterious finding: despite robust expression of MHC II machinery, AT2 cells have a surprisingly limiting capacity to process and present endogenous and exogenous antigens. This introduces the room elephant: why then do AT2 cells express MHC II?

This paper will be of broad interest to immunologists, virologists, clinicians etc. My only suggestion is that the authors devote a paragraph or four to speculate that AT2 cells are up to.

Reviewer #2 (Remarks to the Author):

The authors confirm that type II alveolar epithelial cells (AT2s) express major histocompatibility complex II (MHC II) molecules even if being not considered as professional antigen presenting cells (APC). In contrast to previous studies, they report here the mechanisms regulating this MHCII expression in AT2s, they analysed their antigen presentation capacity both in vitro and in vivo and they demonstrated its impact on protection against respiratory viral disease. Most of this work was performed in the mouse model. The authors used different types of knock-out mice to investigate the role of the transcription factor driving MHCII expression, the role of the interferon-g axis as well as the requirement of different other immune mediators. They further demonstrated that AT2s possess mediators of classical MHCII presentation, and are able to express virally derived intracellular antigens which is a prerequisite for endogenous processing during infections. The authors further demonstrated that the expression of MHCII by AT2 contributes to protection of mice from respiratory viral disease. Finally, they very nicely showed that AT2s, even if able to present antigens via MHCII, have a restrained capacity to do it as only some epitopes are efficiently presented. Even using an in vivo model, they demonstrated a limited capacity of the AT2 to present antigens to CD4+ T lymphocytes.

Even if some previous data on the literature reported on the antigen presenting capacity of AT2s, this is the first manuscript reporting in depth analysis of the mechanisms involved in this antigenic presentation so that the information provided in this manuscript is of utmost importance. Understanding these mechanisms is indispensable to shed light on the mechanisms of induction of CD4+ T cell responses to many pathogens invading first the respiratory tract and quite often the AT2s.

The methodology is sound and well described and the data presented clearly support the discussion and the conclusions made by the authors. The figures are clearly presented and explained.

Minor comment:

An important conclusion of this work is that antigen-specific CD4 T cell responses may be induced in the lung thanks to the AT2s which are able to present antigens to CD4+ T lymphocytes, at least to effector CD4+ T cells. As CD4+ T lymphocytes from the lung parenchyma recirculate, this could perhaps explain at least partially the persistence of antigen-specific CD4+ T lymphocytes in the blood after clearance of an active infection in the lung. The discussion would benefit from adding a brief sentence with reference on the recirculation of CD4+ T lymphocytes from the lung parenchyma.

Françoise Mascart, M.D., PhD

Reviewer #3 (Remarks to the Author):

Currently the role of type II alveolar epithelial cells (AT2) in the adaptive response of the lung is not clearly understood. Previous publications have provided evidence that AT2 cells express features of antigen presenting cells and activate CD4 T cells while others have reported the opposite with evidence for suppression of CD4 T cell function and a tolerogenic role.

In this paper Toulmin et al re-examine this question using a combination of gene-targeted mouse models, T cell lines and in vitro assays. They investigate the capacity of AT2 to present antigen by MHC II molecules and further investigate the possible contribution of this cell type in the adaptive immune response of the lung in virus infection using a Cre driven conditional MHC II ablation strategy controlled by the surfactant protein C promoter to target AT2 cells.

The authors report that AT2 cells (AT2s) constitutively express high levels of MHC II regulated independent of an inflammatory signal and that AT2s possess' components of the intracellular machinery (invariant chain and H2M) for MHC II/peptide complex formation but lack CD80 / CD86 from which they conclude AT2s may have capacity to activate effector but not naïve CD4 T cells. They then test the impact of AT2 MHC II (by conditional ablation) on disease outcome in two models of primary virus lung infection (a surprising choice of model given their previous conclusion). Here they found despite no significant difference in lung virus titre, AT2 MHC II gene targeted mice exhibited a lower survival rate (that did not reach statistical significance) relative to the comparator strain. To explain this effect they propose AT2s have a limited capacity for antigen processing and presentation, despite the presence of machinery for antigen processing and presentation. They provide data that AT2s pulsed with synthetic peptide drive negligible / limited activation of T cell hybridomas or IFN γ production by primed CD4 T cells from virus infected AT2s in ex vivo antigen presenting

assays when cross compared with professional APCs (B cells / CD103+ CD11c+ cells). To decipher the mechanism, in a non-infectious model system (presentation of the E α 52–68 I-Ab self-peptide complex) they infer that inflammatory signals (IFN γ) increase the capacity of AT2s to present MHC II/peptide complexes by a mechanism they suggest is linked to IFN γ driven upregulation of H-2M. From this they conclude AT2s are not deficient in antigen presentation, but suggest MHC II antigen presentation is restrained by an AT2 cell intrinsic mechanism that opposes MHC II presentation. This would be very interesting and novel if confirmed but data in support is not provided.

In sum, this study resolves some of the earlier data and provides a mechanistic insight to clarify previous conflicting data, however the key data to support the important question of how AT2 MHC II contribute to protection against respiratory viral disease is not in my opinion conclusively demonstrated. Further evidence is required. Experimental demonstration of what may 'restrain' MHC II presentation would be important and novel for the field. For example does H2-O moderate H2-DM-mediated peptide loading?

Specific Comments:

1. Data in Fig 2a are presented as evidence AT2s have capacity for active antigen uptake and degradation (antigen processing machinery), however the shift in DQ-Ova for AT2s is not fully convincing. As formal proof that AT2s possess antigen-processing capability, would require demonstration of protease activity using activity based probes or to demonstrate expression of cathepsins at protein level.

2. What is the basis for concluding H2M α 1 is the predominant form of H-2M expressed in mouse AT2s? The PCR primers do not distinguish which Beta chain is expressed and expression by flow is revealed by staining for the alpha beta2 dimer. Can this be clarified?

3. The murine data report a low frequency of AT2 WT mice express CLIP/I-Ab complexes, implying functional H-2M CLIP editing. However in the same animal models a very low frequency of AT2s (WT B6 and BalbXB6 F1) express H2M (both naïve and in the context of an inflammatory signal). How do the authors reconcile these data?

4. Cre-mediated targeting of Iab is known to be unreliable and leads to ablation of MHC II across several cell types. Acknowledging that the authors tried to mitigate by screening mice SPC delta Ab1 mice, the genetic system used to evaluate the impact of loss of AT2 MHCII on lung virus infection is not clean. Since the role of AT2 MHC II in the context of virus infection is a major point of the paper, it would be better to address this by a cleaner alternative model. What is the impact of ablated MHC II by non-haemopoietic cells in the lung of B6 CD45.1 MHC II +/+ to MHC II -/- bone marrow chimeric mice and their controls when infected with IAV or Sendai virus?

5. As discussed above, the paper would benefit from experimental demonstration in support of a mechanism that may 'restrain' MHC II presentation.

Minor comments

1. Fig 1f and g; show the actual MFI values (as opposed to % of WT)

2. Fig 1 d, f, g; Fig 2 c, need to increase n to at least 3 to compute the mean and SEM

3. Supp fig 1: gating strategy for mouse AT2, why is a live/dead gate not included in the strategy?
4. Supp fig 3, show the actual MFI values (as opposed to % of WT)
5. Fig 3 b are there better images for SPC delta Ab1 and an image for the control SPC Cre?
6. Page 11, penultimate line: the text 'MHC II on AT2s serves to reduce morbidity and mortality in the setting of respiratory illness' is not substantiated by the current data and would requires moderation.
7. Similarly, page 15, line 7 - the current data does not substantiate the text in the discussion 'AT2 MHC II improves the outcome of respiratory disease in vivo' .

We thank the reviewers for their careful consideration and for their constructive feedback on our initial manuscript submission. We have included two drafts of the revised manuscript: one with new major changes highlighted in yellow, the other unmarked. We have addressed all comments as follows (concerns and suggestions highlighted in yellow, and responses highlighted in green).

Reviewer #1 (Remarks to the Author):

Toulmin et al. comprehensively characterize the curious expression of MHC class II molecules (MHC II) in type II alveolar cells (AT2). This is a particularly timely study, given the high expression of ACE2, the SARS-CoV-2 receptor, on AT2 cells, and their critical role in COVID 19.

The constitutive expression of MHC II on non-professional APCs is an oddity, and it has been known for some time that MHC II is expressed on AT2. Until this paper, however, the function of these molecules in antigen presentation and protection against lung infections was completely uncharacterized.

In this marvelous study, the authors show that AT2 MHC II expression is regulated in a manner that is independent of IFN- γ , and indeed, even CIITA, the master regulator of MHC II. This finding alone would make the paper worthy of publication in Nat Comm, but the authors go on to provide the first functional characterization of AT2 antigen presentation. They demonstrate that AT2 cells express both invariant chain and functional H2M, which appropriately reduces MHC II CLIP peptide expression.

By selectively knocking out MHC II in AT2 cells in vivo, Toulmin et al. establish that MHC II expression is not required for AT2 function or maintaining baseline lung immunity, but are required for optimal recovery from IAV or Sendai virus infection. This sets the stage for their most important, if mysterious finding: despite robust expression of MHC II machinery, AT2 cells have a surprisingly limiting capacity to process and present endogenous and exogenous antigens. This introduces the room elephant: why then do AT2 cells express MHC II?

This paper will be of broad interest to immunologists, virologists, clinicians etc. **My only suggestion is that the authors devote a paragraph or four to speculate that AT2 cells are up to.**

Reviewer #2 (Remarks to the Author):

The authors confirm that type II alveolar epithelial cells (AT2s) express major histocompatibility complex II (MHC II) molecules even if being not considered as professional antigen presenting cells (APC). In contrast to previous studies, they report here the mechanisms regulating this MHCII expression in AT2s, they analysed their antigen presentation capacity both in vitro and in vivo and they demonstrated its impact on protection against respiratory viral disease. Most of this work was performed in the mouse model. The authors used different types of knock-out mice to investigate the role of the transcription factor driving MHCII expression, the role of the interferon-g axis as well as the requirement of different other immune mediators. They further demonstrated that AT2s possess mediators of classical MHCII presentation, and are able to express virally derived intracellular antigens which is a prerequisite for endogenous processing during infections. The authors further

demonstrated that the expression of MHCII by AT2 contributes to protection of mice from respiratory viral disease. Finally, they very nicely showed that AT2s, even if able to present antigens via MHCII, have a restrained capacity to do it as only some epitopes are efficiently presented. Even using an in vivo model, they demonstrated a limited capacity of the AT2 to present antigens to CD4+ T lymphocytes.

Even if some previous data on the literature reported on the antigen presenting capacity of AT2s, this is the first manuscript reporting in depth analysis of the mechanisms involved in this antigenic presentation so that the information provided in this manuscript is of utmost importance. Understanding these mechanisms is indispensable to shed light on the mechanisms of induction of CD4+ T cell responses to many pathogens invading first the respiratory tract and quite often the AT2s.

The methodology is sound and well described and the data presented clearly support the discussion and the conclusions made by the authors. The figures are clearly presented and explained.

Minor comment:

An important conclusion of this work is that antigen-specific CD4 T cell responses may be induced in the lung thanks to the AT2s which are able to present antigens to CD4+ T lymphocytes, at least to effector CD4+ T cells. As CD4+ T lymphocytes from the lung parenchyma recirculate, this could perhaps explain at least partially the persistence of antigen-specific CD4+ T lymphocytes in the blood after clearance of an active infection in the lung. **The discussion would benefit from adding a brief sentence with reference on the recirculation of CD4+ T lymphocytes from the lung parenchyma.**

Françoise Mascal, M.D., PhD

We thank reviewers #1 and #2 for their enthusiastic assessment of our manuscript, and we appreciate their helpful suggestions to improve the discussion section. Accordingly, we have expanded this section to include a more comprehensive discussion of the possible functions of AT2 MHCII as well as its potential impact on lung CD4+ T cell trafficking, as requested.

Reviewer #3 (Remarks to the Author):

Currently the role of type II alveolar epithelial cells (AT2) in the adaptive response of the lung is not clearly understood. Previous publications have provided evidence that AT2 cells express features of antigen presenting cells and activate CD4 T cells while others have reported the opposite with evidence for suppression of CD4 T cell function and a tolerogenic role.

In this paper Toulmin et al re-examine this question using a combination of gene-targeted mouse models, T cell lines and in vitro assays. They investigate the capacity of AT2 to present antigen by MHC II molecules and further investigate the possible contribution of this cell type in the adaptive immune response of the lung in virus infection using a Cre driven conditional MHC II ablation strategy controlled by the surfactant protein C promoter to target AT2 cells.

The authors report that AT2 cells (AT2s) constitutively express high levels of MHC II regulated independent of an inflammatory signal and that AT2s possess' components of the intracellular machinery (invariant chain and H2M) for MHC II/peptide complex formation but lack CD80 / CD86 from which they conclude AT2s may have capacity to activate effector but not naïve CD4 T cells. They then test the impact of AT2 MHC II (by conditional ablation) on disease outcome in two models of primary virus lung infection (a surprising choice of model given their previous conclusion). Here they found despite no significant difference in lung virus titre, AT2 MHC II gene targeted mice exhibited a lower survival rate (that did not reach statistical significance) relative to the comparator strain. To explain this effect they propose AT2s have a limited capacity for antigen processing and presentation, despite the presence of machinery for antigen processing and presentation. They provide data that AT2s pulsed with synthetic peptide drive negligible / limited activation of T cell hybridomas or IFN γ production by primed CD4 T cells from virus infected AT2s in ex vivo antigen presenting assays when cross compared with professional APCs (B cells / CD103+ CD11c+ cells). To

decipher the mechanism, in a non-infectious model system (presentation of the E α 52–68 I-Ab self-peptide complex) they infer that inflammatory signals (IFN γ) increase the capacity of AT2s to present MHC II/peptide complexes by a mechanism they suggest is linked to IFN γ driven upregulation of H-2M. From this they conclude AT2s are not deficient in antigen presentation, but suggest MHC II antigen presentation is restrained by an AT2 cell intrinsic mechanism that opposes MHC II presentation. This would be very interesting and novel if confirmed but data in support is not provided.

In sum, this study resolves some of the earlier data and provides a mechanistic insight to clarify previous conflicting data, however the key data to support the important question of how AT2 MHC II contribute to protection against respiratory viral disease is not in my opinion conclusively demonstrated. Further evidence is required. Experimental demonstration of what may 'restrain' MHC II presentation would be important and novel for the field. For example does H2-O moderate H2-DM-mediated peptide loading?

Specific Comments:

1. Data in Fig 2a are presented as evidence AT2s have capacity for active antigen uptake and degradation (antigen processing machinery), however the shift in DQ-Ova for AT2s is not fully convincing. As formal proof that AT2s possess antigen-processing capability, would require demonstration of protease activity using activity based probes or to demonstrate expression of cathepsins at protein level.

We feel the DQ-Ova experiment makes a strong case. The addition of the DQ-Ova substrate at 37C results in a histogram that is essentially non-overlapping with the 4C counterpart. This is accompanied by an MFI change from 8325 to 31658. While the overall magnitude of this MFI shift (3.8x) is lower than that of DCs (12x), it is similar to that of B cells (4.7x).

Nonetheless, we agree that the DQ-Ova system assesses two processes simultaneously (both antigen uptake and degradation), and a more straightforward approach to assess degradative capacity alone would be to evaluate protease activity. To address this, we have added studies directly demonstrating activity of two cathepsin enzymes-- Cathepsin D and L -- in AT2s to Supplemental Figure 5. We have also added an RNA-sequencing analysis to the same figure that demonstrates transcript expression for a wide variety of cathepsins, as well as the antigen processing enzymes GILT and AEP, in AT2s.

2. What is the basis for concluding H2M α 1 is the predominant form of H-2M expressed in mouse AT2s? The PCR primers do not distinguish which Beta chain is expressed and expression by flow is revealed by staining for the alpha beta2 dimer. Can this be clarified?

We concluded that H2M α 1 is the predominant isoform of H2M in AT2s by synthesizing multiple complementary pieces of data, all demonstrated in Figure 2. AT2s exhibit complete unloading of CLIP from MHCII in WT mice, but not in H2M $^{-/-}$ mice. Thus, WT AT2s must express functioning H2M protein. In theory, this H2M protein could be either isoform of H2M -- H2M α 1 or H2M α 2. By flow cytometry, AT2s do not express the H2M α 2 isoform (using an antibody specific for the H2M α 2 isoform that does not cross-react with H2M α 1). Thus, as AT2s express functional H2M protein, but it is not H2M α 2, we conclude that the H2M isoform they express is H2M α 1. We have modified the text to further clarify this. We have also added an RNA-sequencing analysis to Figure S5 that more directly demonstrates the expression of H2M α and the predominance of H2M β 1 over H2M β 2 in AT2s.

3. The murine data report a low frequency of AT2 WT mice express CLIP/I-Ab complexes, implying functional H-2M CLIP editing. However in the same animal models a very low frequency of AT2s (WT B6 and BalbXB6 F1) express H2M (both naïve and in the context of an inflammatory signal). How do the authors reconcile these data?

H2M can exist in two forms- H2M $\alpha\beta$ 1 or H2M $\alpha\beta$ 2. Both forms are capable of mediating CLIP unloading and peptide exchange (see manuscript reference #49). Our data demonstrate that a low frequency of AT2s in F1 mice express the H2M $\alpha\beta$ 2 isoform of H2M specifically, similarly to B6 WT AT2s as shown in Figure 2. Note that the antibody used in these studies does not cross react with H2M $\alpha\beta$ 1, so the lack of staining does not indicate an absence of all isoforms of H2M – it only demonstrates lack of H2M $\alpha\beta$ 2. Intact H2M-mediated CLIP editing with absent H2M $\alpha\beta$ 2 is consistent with expression of H2M restricted to just the H2M $\alpha\beta$ 1 isoform. We have modified the text to further clarify this.

4. Cre-mediated targeting of I-Ab is known to be unreliable and leads to ablation of MHC II across several cell types. Acknowledging that the authors tried to mitigate by screening mice SPC delta Ab1 mice, the genetic system used to evaluate the impact of loss of AT2 MHCII on lung virus infection is not clean. Since the role of AT2 MHC II in the context of virus infection is a major point of the paper, it would be better to address this by a cleaner alternative model. What is the impact of ablated MHC II by non-haemopoietic cells in the lung of B6 CD45.1 MHC II +/+ to MHC II-/- bone marrow chimeric mice and their controls when infected with IAV or Sendai virus?

We acknowledge the caveats of the Cre + floxed I-Ab system, which we address in the manuscript. These include (1) the possibility of germline recombination, which we avoided by screening all mice before use (described in Methods), and (2) that the presence of the floxed MHCII locus (regardless of Cre) leads to lower levels of MHCII expression in some cell types, as we demonstrate in Figure 3. We addressed this 2nd caveat by using floxed-IAb-only mice as controls, instead of Cre-only, so that both the experimental and control mice we compared were subject to this same effect equally.

With both these caveats addressed, the Cre + floxed I-Ab system is by far the cleanest system to answer our question. We considered using bone marrow chimeras early in our studies but decided against this for the following reasons. 1) Using bone marrow chimeras requires substantial manipulation of the mice, which introduces several additional and unnecessary experimental variables that are more difficult to control, including the effects of radiation, antibiotic treatment, transient immunocompromised state, and reconstitution success. 2) Furthermore, creation of mice with MHCII ablated on all nonhematopoietic cells would require transfer of WT bone marrow into a MHCII-/- host (WT \rightarrow MHCII-/-). These mice would lack MHCII on thymic epithelial cells and thus would have absent positive CD4⁺ T cell selection, resulting in the failure of CD4⁺ T cell development. 3) Even if these mice received an adoptive transfer of CD4⁺ T cells prior to infection in an attempt to overcome this defect, this would likely not approximate the full CD4⁺ T cell repertoire in the control WT \rightarrow WT chimeric mice, adding an additional confounding variable to the interpretation of the experiment. 4) Finally, in this bone marrow chimera system all nonhematopoietic cells would lose MHCII, not just AT2s. As other nonhematopoietic cell types in the lung express MHCII, such as endothelial cells, this would represent a less clean way to assess the contribution of just AT2 cell MHCII, which is our main question.

5. As discussed above, the paper would benefit from experimental demonstration in support of a mechanism that may 'restrain' MHC II presentation.

In this manuscript, we evaluate several potential mechanistic explanations for restrained antigen presentation by AT2s. We directly evaluate antigen uptake and source protein burden (in both the viral and Yae systems), proteolysis (via DQ-ova, cathepsin assays), and expression and function of the classical MHCII-associated chaperone proteins H2M and invariant chain. The one other MHCII-associated protein that is coregulated with MHCII expression, apart from H2M and invariant chain, is H2O (the H2M antagonist). As mentioned, H2O was not examined in our initial submission. Thus, in this revised version, we have included an analysis of H2O expression by AT2s, in Supplemental Figure 5. We have also added an RNA-sequencing analysis to Figure S5 that demonstrates expression of all conventional MHCII-associated genes, as well as many antigen processing enzymes, in AT2s.

Although we were unable to identify one precise factor underlying inefficient AT2 MHCII antigen presentation, we did rule out all the most likely explanations, as discussed above. While we very much agree that further studies pinpointing the specific mechanism of poor/restrained MHCII presentation by AT2s would be of great interest, we believe that additional studies beyond those we already included in the manuscript would be beyond the scope of this paper and would constitute a large scale and time-intensive undertaking.

Minor comments

1. Fig 1f and g; show the actual MFI values (as opposed to % of WT)

These data were acquired across several different flow cytometry experiments – not all KO mouse lungs were assessed at the same time. As the MFI is subject to experiment-to-experiment variation depending on the cytometer and fluorophores used and associated voltages/gains, comparing raw MFI values between these experiments would not be appropriate here. However, in each individual experiment performed, WT and MHCII^{-/-} control mice were included as a reference for the knockout mice of interest. Normalizing the MFI of AT2s from KO mice to the MFI of AT2s from WT mice acquired within the same given experiment is a more suitable way to compare MHCII expression between KO strains across separate experiments.

2. Fig 1 d, f, g; Fig 2 c, need to increase n to at least 3 to compute the mean and SEM

While we agree that generally a larger n is preferable, we were limited in some cases to n=2 based on the availability of some of the genetic knockout mice. Nonetheless, the mean and SEM calculations are still valid with n=2, and for our studies, the error calculations were performed automatically based on the input data using GraphPad Prism software. Please see: <https://www.graphpad.com/support/faqid/591/>

3. Supp fig 1: gating strategy for mouse AT2, why is a live/dead gate not included in the strategy?

This gating strategy reflects the strategy used for cell sorting. Live/dead cell staining was not used during any cell sorting experiments to minimize sample processing time to preserve cell

viability, as discussed in the methods section. The text in the figure legend has been modified to clarify this.

4. Supp fig 3, show the actual MFI values (as opposed to % of WT)

See explanation for Minor Comment 1.

5. Fig 3 b are there better images for SPC delta Ab1 and an image for the control SPC Cre?

We are unclear on the aspects of the images that need to be improved. There are other possible images, but they all look very similar. We have modified the language in the figure legend to reflect that these images are representative. There are no images for the SPC-Cre only controls (they were not included in the immunofluorescence analyses) since this strain was not used in any of the *in vivo* experiments.

6. Page 11, penultimate line: the text 'MHC II on AT2s serves to reduce morbidity and mortality in the setting of respiratory illness' is not substantiated by the current data and would require moderation.

The text has been modified to further qualify the language. We have also changed the title of the paper to be more consistent with the modified text.

7. Similarly, page 15, line 7 - the current data does not substantiate the text in the discussion 'AT2 MHC II improves the outcome of respiratory disease *in vivo*' .

The text has been modified to further qualify the language. We have also changed the title of the paper to be more consistent with the modified text.

In addition to the modifications specifically recommended by reviewers #1-3, we also made a few changes that we feel strengthen the study and manuscript overall. Although not explicitly requested, we modified the title of the paper to reflect more qualified language based on the comments of reviewer #3 that the *in vivo* findings were overstated. Additionally, we replaced the homeostasis analysis of 7-month-old mice with an analogous analysis of 12-week-old mice; the results of the analysis are very similar, but we felt that using 12-week-old mice would ensure better consistency and comparability with the other *in vivo* infection experiments, which were done with 8-12 week old mice.

REVIEWER COMMENTS

Reviewer #1:

Remarks to the Author:

The authors have done an exemplary job in positively responding to the reviews. I completely agree with their arguments not to perform the bone marrow chimera experiments suggested by reviewer 3. These are notoriously difficult and unreliable. We stopped doing such experiments years ago having learned our lesson!

Reviewer #2:

Remarks to the Author:

The authors significantly improved the discussion concerning the generation of memory CD4+ T lymphocytes as recommended as minor comment.

Françoise Mascart, M.D., PhD.

Reviewer #3:

Remarks to the Author:

REVIEWER COMMENTS

The authors have provided considered answers in their response, some additional data and have overall added clarity to points raised in the revised manuscript.

There are nonetheless a few points which remain to be addressed which I feel would strengthen the manuscript:

1. The authors “replaced the homeostasis analysis of 7-month-old mice with an analogous analysis of 12-week-old mice” which are more comparable in age to mice used in the infection experiments. This new data set indicates that the frequency of PD1+ CD4+ and PD1+ CD8+ T cells is significantly lower in the lungs of the MHC II deficient SPC delta Ab1 mice when cross compared with the Abfl/fl control group, $p=0.003$ (fig S6a). From this data set one cannot conclude AT2 MHCII expression is dispensable for normal immune homeostasis in the lung, nor that expression is ‘similar’ (page 10). The key point is that the data suggests AT2 MHCII directly / indirectly impairs PD1 expression in this system. Given that respiratory epithelial cells express PDL-1 and expression is increased by inflammatory signals and virus infection, the relative frequency of PD1+ T cells in the two stains of mice may equally explain lower morbidity and mortality in the MHC II deficient SPC delta Ab1 mice. Increased PD1-mediated suppression of effector T cells in the Abfl/fl control group control may ‘restrain’ immunopathology and contribute to improved disease outcome. Thus, while I agree AT2 cells provide a check point on effector T cells in the lung, the new data implies that the mechanism is more complex than the ‘restraint’ in AT2 antigen presentation emphasised in this manuscript and this needs to be modified. Can the authors demonstrate if there is there an expansion of PD1+ CD4+ and PD1+ CD8+ T cells in the lungs of IAV or SeV infected Abfl/fl mice relative to SPC delta Ab1 mice? Secondly do AT2 cells upregulate PDL-1 expression in these virus infected mice? I think concluding that MHC II expression restrains Ag presentation in AT2 cells overtly simplifies the biology of what is going on in the context of virus infection.

2. Minor point relating to the new lung data (Fig S6 and fig 3f-i) is there a typo or was the experimental design changed, as not clear if vascular CD4 and CD8 T cells are excluded.

3. REVIEWERS COMMENT

Fig 1 d, f, g:, Fig 2 c, need to increase n to at least 3 to compute the mean and SEM

3. AUTHORS RESPONSE:

While we agree that generally a larger n is preferable, we were limited in some cases to n=2 based on the availability of some of the genetic knockout mice. Nonetheless, the mean and SEM calculations are still valid with n=2, and for our studies, the error calculations were performed automatically based on the input data using GraphPad Prism software.

3. REVIEWERS COMMENT: While that may be true from the perspective of GraphPad Prism, n=2 is really small and may result in overconfidence in findings obtained in small-sample studies. Do the authors really think n=2 is rigorous in genetic-knockout mice ?

4. Page 8, I think the H2-O data in fig S5d (and not as stated in the text)

5. Page 9 sub heading, change to 'protection' to improved respiratory disease outcome'

We thank the reviewers again for their thoughtful consideration of our revised manuscript submission. As before, we have included two drafts of the manuscript: one with new major and requested changes highlighted in yellow, the other unmarked. Below we have addressed all comments – verbatim reviewer comments are in dark gray text and our responses are bulleted in black text for ease of reading.

REVIEWER COMMENTS

Reviewer #1 (Remarks to the Author):

The authors have done an exemplary job in positively responding to the reviews. I completely agree with their arguments not to perform the bone marrow chimera experiments suggested by reviewer 3. These are notoriously difficult and unreliable. We stopped doing such experiments years ago having learned our lesson!

Reviewer #2 (Remarks to the Author):

The authors significantly improved the discussion concerning the generation of memory CD4+ T lymphocytes as recommended as minor comment.
Françoise Mascart, M.D., PhD.

- We thank both reviewers #1 and #2 for their continued enthusiasm for our manuscript and appreciate their encouraging remarks.

Reviewer #3 (Remarks to the Author):

REVIEWER COMMENTS

The authors have provided considered answers in their response, some additional data and have overall added clarity to points raised in the revised manuscript.

There are nonetheless a few points which remain to be addressed which I feel would strengthen the manuscript:

- We thank reviewer #3 for their feedback. Below we address each comment on a point-by-point basis.

1. The authors “replaced the homeostasis analysis of 7-month-old mice with an analogous analysis of 12-week-old mice” which are more comparable in age to mice used in the infection experiments. This new data set indicates that the frequency of PD1+ CD4+ and PD1+ CD8+ T cells is significantly lower in the lungs of the MHC II deficient SPC delta Ab1 mice when cross compared with the Abfl/fl control group, $p=0.003$ (fig S6a). From this data set one cannot conclude AT2 MHCII expression is dispensable for normal immune homeostasis in the lung, nor that expression is ‘similar’ (page 10). The key point is that the data suggests AT2 MHCII directly / indirectly impairs PD1 expression in this system. Given that respiratory epithelial cells express PDL-1 and expression is increased by inflammatory signals and virus infection, the relative frequency of PD1+ T cells in the two stains of mice may equally explain lower morbidity and mortality in the MHC II deficient SPC delta Ab1 mice. Increased PD1-mediated suppression of effector T cells in the Abfl/fl control group control may

'restrain' immunopathology and contribute to improved disease outcome. Thus, while I agree AT2 cells provide a check point on effector T cells in the lung, the new data implies that the mechanism is more complex than the 'restraint' in AT2 antigen presentation emphasised in this manuscript and this needs to be modified. Can the authors demonstrate if there is there an expansion of PD1+ CD4+ and PD1+ CD8+ T cells in the lungs of IAV or SeV infected Ab1/fl mice relative to SPC delta Ab1 mice? Secondly do AT2 cells upregulate PDL-1 expression in these virus infected mice? I think concluding that MHC II expression restrains Ag presentation in AT2 cells overtly simplifies the biology of what is going on in the context of virus infection.

- We have modified our description of the homeostasis data in the text to more clearly point out the comparisons that yielded $P < 0.05$.
- We have now included studies measuring PD1 expression by T cells as well as PD-L1 expression by AT2s after IAV infection, shown in Supplemental Figure 9. In these studies, we have also included analyses of the other subsets for which there were minor yet statistically significant differences at homeostasis.
- Based on the reviewer's comments above, including "while I agree AT2 cells provide a check point on effector T cells in the lung, the new data implies that the mechanism is more complex than the 'restraint' in AT2 antigen presentation emphasised in this manuscript", we believe there is some confusion regarding our assertions about how AT2 MHCII may contribute to differences in respiratory viral disease outcome. We wish to clarify our conclusions here.
 - a. We do not state at any point in the text that the mechanism underlying differences in respiratory viral disease between SPC Δ Ab1 and Ab1^{fl/fl} mice is the *restraint* in AT2 MHCII antigen presentation. Instead, in the paper we make the two following and distinct conclusions: (1) the presence or absence of MHCII on AT2s leads to a difference in outcome between SPC Δ Ab1 and Ab1^{fl/fl} mice, and separately (2) AT2s exhibit limited MHCII antigen presentation capacity. From our studies in SPC Δ Ab1 and Ab1^{fl/fl} mice, we can and do only conclude that MHCII expression/presentation by AT2s leads to a difference in respiratory viral disease outcome in mice. To specifically ask whether the *restraint* in AT2 MHCII presentation is important would require a different study altogether. We have further revised the title of the paper in order to clarify this.
 - b. At the beginning of the final results subsection, we do hypothesize that a limitation in AT2 MHCII antigen presentation may explain why we did not observe a bigger difference between SPC Δ Ab1 and Ab1^{fl/fl} mortality or weight loss. This is offered as a possible explanation for why the difference we observed was relatively moderate in magnitude; it is not an explanation for the difference itself, which we can only conclude is due to the presence or absence of MHCII on AT2s. We initially also had a half sentence in this paragraph that speculated that restrained MHCII presentation by AT2s might be beneficial in the lung in general, but this has been removed to prevent confusion.
 - c. In the discussion, we speculate as to why AT2s may have evolved to possess limited antigen presentation capacity despite expressing such a high level of MHCII protein—that in general in the lung it may be beneficial for AT2s to present via MHCII in a restrained manner rather than a robust one, so that they may contribute to

responses in a measured fashion without overamplifying T cell-mediated inflammation. This is not offered as an explanation for the differences in outcome between SPC^{ΔAb1} and Ab1^{fl/fl} mice, but rather is a hypothetical reconciliation of our two observations that (1) AT2 MHCII presentation seems to be beneficial in respiratory viral disease outcome, yet (2) AT2 MHCII presentation occurs in a restricted manner compared to professional APCs.

2. Minor point relating to the new lung data (Fig S6 and fig 3f-i) is there a typo or was the experimental design changed, as not clear if vascular CD4 and CD8 T cells are excluded.

- Vascular cells are not excluded in the lung data.

3. REVIEWERS COMMENT

Fig 1 d, f, g; Fig 2 c, need to increase n to at least 3 to compute the mean and SEM

3. AUTHORS RESPONSE:

While we agree that generally a larger n is preferable, we were limited in some cases to n=2 based on the availability of some of the genetic knockout mice. Nonetheless, the mean and SEM calculations are still valid with n=2, and for our studies, the error calculations were performed automatically based on the input data using GraphPad Prism software.

3. REVIEWERS COMMENT: While that may be true from the perspective of GraphPad Prism, n=2 is really small and may result in overconfidence in findings obtained in small-sample studies. Do the authors really think n=2 is rigorous in genetic-knockout mice ?

- We have increased the samples sizes to n=5 for all strains in figures 1f and 1g. For figures 1d and 2c (qPCR studies), the DC sample in each graph remains as n=2 biological replicates, composed of 3 technical replicates per measurement. As we have RNA-sequencing analyses of the same genes (fig S5a) with n=5 biological replicates that directly corroborates these qPCR findings, we are confident in these data.

4. Page 8, I think the H2-O data in fig S5d (and not as stated in the text)

- The text has been modified accordingly.

5. Page 9 sub heading, change to 'protection' to improved respiratory disease outcome'

- The text has been modified accordingly.

REVIEWER COMMENTS

Reviewer #3 (Remarks to the Author):

The authors have provided additional data and made changes to the revised manuscript that are generally satisfactory and have address my concerns. I would however like them to amend the text of the following sentences that relate to their reporting of the difference in frequency of lung CD4+ PD1+ T cells they observe between SPC delta Ab1 and Ab1fl/fl mice. Looking at the data (fig S6a) the differences are not “minor” as stated. Thus, please change the wording to delete the word “minor”, "minor yet" and “slight” from each of the following sentences:

Page 10, line 22-23: “Minor” differences between SPC delta Ab1 and Ab1fl/fl mice achieve statistical significance in the cases of CD4+ T cell numbers,..... the proportion of CD4s expressing the activation / exhaustion marker PD1.

Page 12, final paragraph: We also asked whether the “minor, yet” statistically significant differences between certain lung T cell subsets in SPC delta Ab1 and Ab1fl/fl mice

Page 13, line 5, Thus, the “slight” differences in magnitude of these T cell subsets at homeostasis are unlikely to explain the differences in outcomes following infection.

We thank reviewer 3 again for their feedback. Below we have addressed all comments – verbatim reviewer comments are in dark gray text and our responses are bulleted in black text for ease of reading.

REVIEWER COMMENTS

Reviewer #1 (Remarks to the Author):

The authors have provided additional data and made changes to the revised manuscript that are generally satisfactory and have address my concerns. I would however like them to amend the text of the following sentences that relate to their reporting of the difference in frequency of lung CD4+ PD1+ T cells they observe between SPC delta Ab1 and Ab1fl/fl mice. Looking at the data (fig S6a) the differences are not “minor” as stated. Thus, please change the wording to delete the word “minor”, "minor yet" and “slight” from each of the following sentences:

Page 10, line 22-23: “Minor” differences between SPC delta Ab1 and Ab1fl/fl mice achieve statistical significance in the cases of CD4+ T cell numbers,..... the proportion of CD4s expressing the activation / exhaustion marker PD1.

- The text has been modified accordingly.

Page 12, final paragraph: We also asked whether the “minor, yet” statistically significant differences between certain lung T cell subsets in SPC delta Ab1 and Ab1fl/fl mice

- The text has been modified accordingly.

Page 13, line 5, Thus, the “slight”differences in magnitude of these T cell subsets at homeostasis are unlikely to explain the differences in outcomes following infection.

- The text has been modified accordingly.